# STAT1 potentiates oxidative stress revealing a targetable vulnerability that increases phenformin efficacy in breast cancer

Stephanie P. Totten[1,2], Young Kyuen Im[1,2], Eduardo Cepeda Cañedo[1], Ouafa Najyb[3,4], Alice Nguyen [1], Steven Hébert [1], Ryuhjin Ahn [1,2], Kyle Lewis [1,2], Benjamin Lebeau[1,2], Rachel La Selva[1,2], Valérie Sabourin[1], Constanza Martínez[4], Paul Savage [4], Hellen Kuasne[4], Daina Avizonis[4], Nancy Santos Martínez [1], Catherine Chabot[1], Adriana Aguilar-Mahecha[1], Marie-Line Goulet[1], Matthew Dankner[2,4], Michael Witcher [1,2,5], Kevin Petrecca [6], Mark Basik[1,2,5], Michael Pollak[1,2,5], Ivan Topisirovic [1,2,3,5], Rongtuan Lin[1,7], Peter M. Siegel [2,3,4], Claudia L. Kleinman [1,8], Morag Park [3,4,5], Julie St-Pierre [9] & Josie Ursini-Siegel [1,2,3,5✉]

Bioenergetic perturbations driving neoplastic growth increase the production of reactive oxygen species (ROS), requiring a compensatory increase in ROS scavengers to limit oxidative stress. Intervention strategies that simultaneously induce energetic and oxidative stress therefore have therapeutic potential. Phenformin is a mitochondrial complex I inhibitor that induces bioenergetic stress. We now demonstrate that inflammatory mediators, including IFNγ and polyIC, potentiate the cytotoxicity of phenformin by inducing a parallel increase in oxidative stress through STAT1-dependent mechanisms. Indeed, STAT1 signaling downregulates NQO1, a key ROS scavenger, in many breast cancer models. Moreover, genetic ablation or pharmacological inhibition of NQO1 using β-lapachone (an NQO1 bioactivatable drug) increases oxidative stress to selectively sensitize breast cancer models, including patient derived xenografts of HER2+ and triple negative disease, to the tumoricidal effects of phenformin. We provide evidence that therapies targeting ROS scavengers increase the anti-neoplastic efficacy of mitochondrial complex I inhibitors in breast cancer.

[1] Lady Davis Institute for Medical Research, Jewish General Hospital, Montreal, QC, Canada. [2] Division of Experimental Medicine, McGill University, Montreal, QC, Canada. [3] Department of Biochemistry, McGill University, Montreal, QC, Canada. [4] Goodman Cancer Research Centre, McGill University, Montreal, QC, Canada. [5] Gerald Bronfman Department of Oncology, McGill University, Montreal, QC, Canada. [6] Department of Neurology and Neurosurgery, McGill University, Montreal, QC, Canada. [7] Department of Microbiology and Immunology, McGill University, Montreal, Quebec, Canada. [8] Department Human Genetics, McGill University, Montreal, QC, Canada. [9] Department of Biochemistry, Microbiology and Immunology, Ottawa Institute of Systems Biology, University of Ottawa, Ottawa, ON, Canada. ✉email: giuseppina.ursini-siegel@mcgill.ca

C urrent treatments often fail to control aggressive breast cancers, underscoring the need to identify therapeutic approaches that elicit durable responses with minimal toxicity. This challenge is further compounded by the high degree of tumor heterogeneity in breast cancer[1]. Identification of essential vulnerabilities that distinguish normal and malignant cells would provide opportunities to selectively target such heterogeneous tumors. Metabolic perturbations represent one such vulnerability as cancer cells must engage diverse metabolic pathways to meet their energetic and biosynthetic demands, while simultaneously maintaining redox balance[2].

Biguanides, including metformin and phenformin, suppress mitochondrial ATP production by inhibiting complex I of the electron transport chain[3–5]. The repurposing of biguanides as anticancer agents has gained interest and numerous preclinical studies have demonstrated their therapeutic potential in oncology[6]. However, epidemiological studies examining the effects of metformin on breast cancer incidence have yielded conflicting data[7–9]. Moreover, phase II clinical trials examining the ability of metformin to improve survival in women with breast cancer have been disappointing[10]. Nevertheless, a subset of clinical trials offers insight into the therapeutic potential of biguanides, substantiating their indirect effects on decreasing circulating insulin and glucose levels, leading to reduced insulin receptor signaling in breast tumors[11–13].

The lack of durable responses with biguanides in clinical trials can be explained, in part, by the fact that preclinical studies use significantly higher drug concentrations than can be achieved clinically[14,15]. Moreover, metabolic flexibility and nutrient availability in the tumor microenvironment may contribute to the poor efficacy of biguanides as single anticancer agents[16–18]. Rational combination strategies that lower the concentrations of biguanides needed for antineoplastic activity may revitalize the therapeutic potential of this drug class in oncology[19].

Several studies suggest that metformin may also elicit antitumor immune responses. Metformin acts on CD8[+] T cells to potentiate their effector functions and elicit a memory phenotype[20,21]. Metformin also inhibits the immunosuppressive properties of myeloid-derived suppressor cells in cancer models[22]. Finally, metformin may induce programmed cell death ligand 1 (PD-L1) proteolytic degradation to improve the efficacy of PD-L1/programmed death-1 (PD1) inhibitors[23].

We explored the underappreciated role for complex I inhibitors, including phenformin, as mitochondrial reactive oxygen species (ROS) generators[4,24]. Breast tumor cells upregulating their antioxidant machinery have a survival advantage against therapy-induced oxidative damage[25]. We were interested in identifying strategies to bypass these ROS defense mechanisms, simultaneously sensitizing them to energetic and oxidative stress induced by phenformin. Signal transducer and activator of transcription 1 (STAT1) signaling potentiates antitumor immune responses downstream of type I (IFNα/β) and II (IFNγ) interferons (IFNs), both within tumor cells and immune cells[26,27]. We now show that STAT1 also perturbs the antioxidant defense mechanisms in breast tumors. Increased STAT1 function potentiates oxidative stress in breast cancers, profoundly sensitizing them to the antitumorigenic effects of phenformin. Moreover, combination strategies that block essential ROS scavengers have the potential to increase the clinical impact of this biguanide, and potentially other complex I inhibitors, in oncology.

## Results

### IFNγ-driven STAT1 activation sensitizes breast cancer cells to phenformin in vitro.
Tyrosine kinase inhibitors reduce the metabolic flexibility of cancer cells, sensitizing them to

biguanides[19]. We recently demonstrated that tyrosine kinases engage the ShcA adaptor protein to increase the metabolic flexibility of breast cancer cells[28]. Loss of phospho-tyrosine-dependent ShcA signaling (Y239/240/313F) increased the reliance of breast cancer cells on mitochondrial metabolism, exposing a selective vulnerability to phenformin[28]. Using cell lines established from polyoma virus MT (MT)-driven mammary tumors, we now show that a non-phosphorylatable ShcA mutant (Y313F) sensitizes breast cancer cells to phenformin (Supplementary Figure 1a). Coupled with our observations that loss of pY313-dependent ShcA signaling increases STAT1 expression in breast cancer cells (Supplementary Figure 1b)[29], we considered the possibility that STAT1 may increase phenformin sensitivity. IFNγ is an inflammatory cytokine that activates the STAT1 pathway[26,27]. We selected a low concentration of IFNγ (1 ng/mL) that induces STAT1 transcriptional responses ($\beta2M$, $Tap1$) (Supplementary Figure 1c) but does not impair breast cancer cell growth (Fig. 1a). IFNγ-induced STAT1 activation cooperates with phenformin to reduce the viability of MT breast cancer cells (Fig. 1a). In contrast, STAT1-overexpressing MT/313F cells show increased phenformin sensitivity, which cannot be further potentiated by IFNγ co-treatment (Fig. 1a). IFNγ and phenformin also cooperatively elicit antitumorigenic responses in murine ErbB2[+] (NOP6, NOP23, and NIC) (Fig. 1b) and human breast cancer cell lines representative of ER[+]/HER2[−] (MCF7), HER2[+] (HCC1954, BT474), and triple-negative (MDA-MB-231, BT20, MDA-MB-436, Hs578T, and BT549) disease (Fig. 1c). These results demonstrate that IFNγ increases the antitumorigenic effects of phenformin in multiple breast cancer models.

To assess whether STAT1 is required for this observed increase in treatment sensitivity, we deleted STAT1 from two MT-transformed cell lines (864 and 4788) using CRISPR/Cas9 genomic editing (Fig. 1d)[29]. As expected, STAT1-deficient cells were unable to upregulate STAT1 target genes ($Irf9$ and $Psmb8$) (Supplementary Figure 1d) following IFNγ stimulation. Using these STAT1-null cells, we found that STAT1 is required for IFNγ to sensitize breast cancer cells to the antitumorigenic effects of phenformin (Fig. 1e). Thus, IFNγ sensitizes multiple breast cancer cell lines, spanning distinct molecular subtypes, to phenformin in a STAT1-dependent manner.

### IFNγ and polyIC sensitize breast tumors to the tumoricidal effects of phenformin in vivo in a STAT1-dependent manner.
We next asked whether increased inflammatory responses also sensitized mammary tumors to the antineoplastic effects of phenformin in vivo. MT4788 cells were injected into the mammary fat pads of IFNγ[+/+] and IFNγ[−/−] mice, both on a pure FVB background[29]. When tumors reached 150 mm[3], mice received daily intraperitoneal injections with 50 mg/kg phenformin or phosphate-buffered saline (PBS) as the vehicle control. Phenformin led to a 30% reduction in tumor growth in IFNγ[+/+], and not IFNγ[−/−], animals (Fig. 2a). As IFNγ is required for tumor immune surveillance, we assessed the relative importance of IFNγ-driven antitumor immunity in conferring increased phenformin sensitivity. We engrafted MT4788 breast cancer cells into the mammary fat pads of CD8[+/+] or CD8[−/−] mice (also on a pure FVB background) and treated mammary tumors with PBS or phenformin (50 mg/kg daily). Although CD8[−/−] mice lack cytotoxic T lymphocytes, we only observed a partial loss in sensitivity of tumors to phenformin, compared to CD8[+/+] controls (Fig. 2b). We also explored the ability of phenformin to enhance the efficacy of therapies that alleviate immune suppression. Tumor-bearing mice were treated with phenformin and a PD1-neutralizing antibody (or isotype control), alone or in combination. Tumor growth was inhibited by 30% in response to either

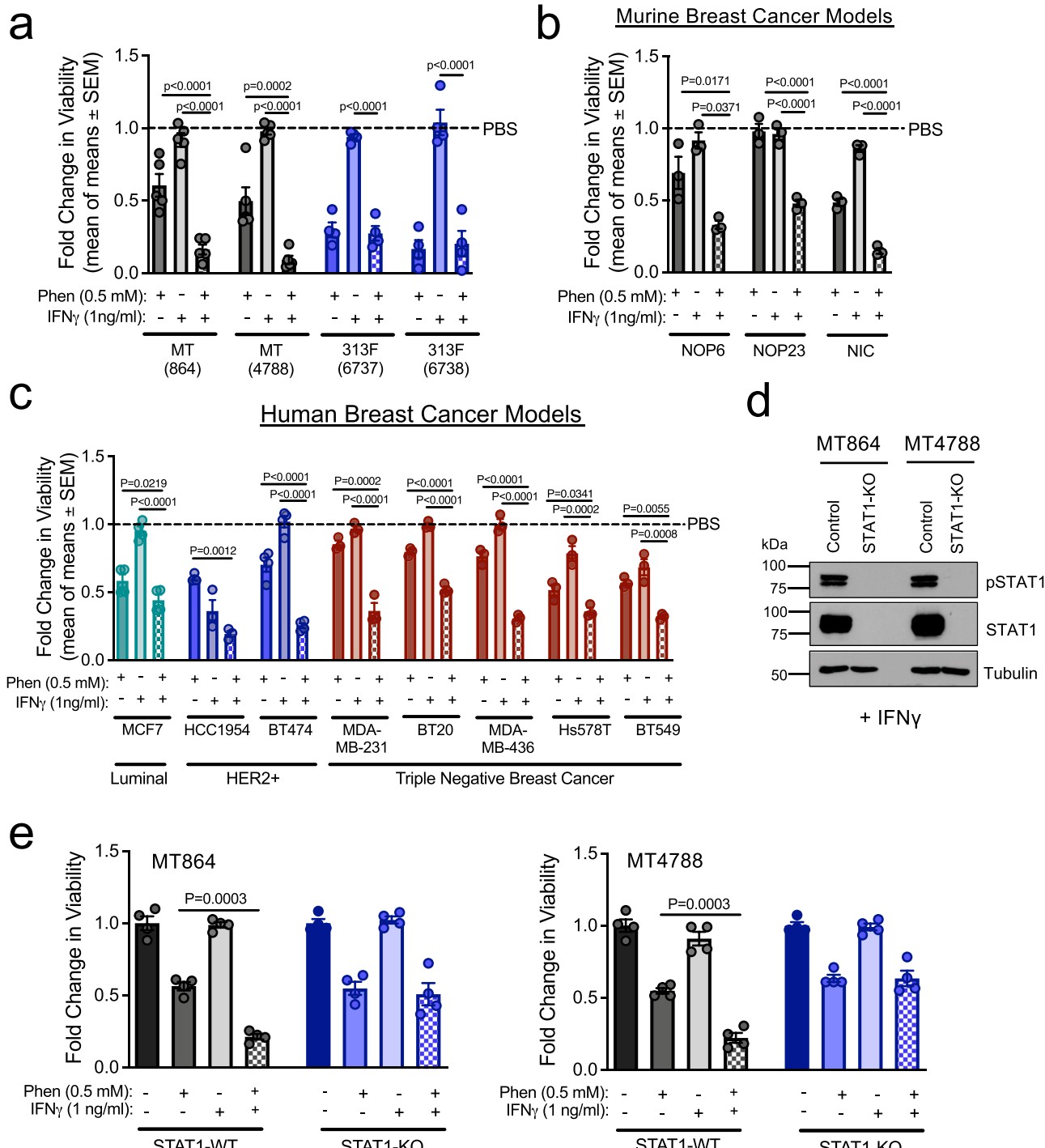

**Fig. 1 IFNγ-driven STAT1 activation sensitizes breast cancer cells to phenformin. a–c** Viability of breast cancer cell lines after treatment with phenformin and IFNγ alone or in combination for 48 h. Data are shown as a fold change in viability compared to PBS controls (**a**, **b**) murine **a** MT864, MT4788: n = 5; 313F 6737 and 6738: n = 4 independent experiments; **b** NOP6, NOP23, NIC: n = 3 independent experiments. **c** Human breast cancer cell lines, MCF7 and BT474: n = 4, the others are n = 3 independent experiments. Data are presented as mean of means ± SEM. **d** Immunoblot analysis of IFNγ-treated STAT1-WT and STAT1-KO cells, representative of n = 3 independent experiments. **e** Viability of MT864 and MT4788-STAT1-WT and STAT1-KO cells treated with PBS, phenformin, or IFNγ, alone or in combination, for 48 h. Data are represented as a fold change in cell viability compared to PBS controls. The graphed data represent one experiment with four technical replicates (mean ± SD) and is representative of two independent experiments. P values were calculated using two-way ANOVA with a Tukey's post hoc test (**a**, **b**, **c**, **e**), and can be found in the figure. See also Supplementary Figure 1.

phenformin or an anti-PD1 antibody alone and an additive effect was observed in combination (Supplementary Figure 1a). We also tested if phenformin could augment the efficacy of the VSV-M (Δ51) oncolytic virus, which kills tumor cells, in part, by augmenting antitumor immune responses[30]. While VSV-M(Δ51) or

phenformin treatment alone elicited antitumor effects in vivo, a further decrease in tumor growth was not observed in mice receiving the combination therapy (Supplementary Figure 2b). Coupled with our in vitro studies, these data suggest that IFNγ-induced phenformin sensitivity does not predominantly rely on

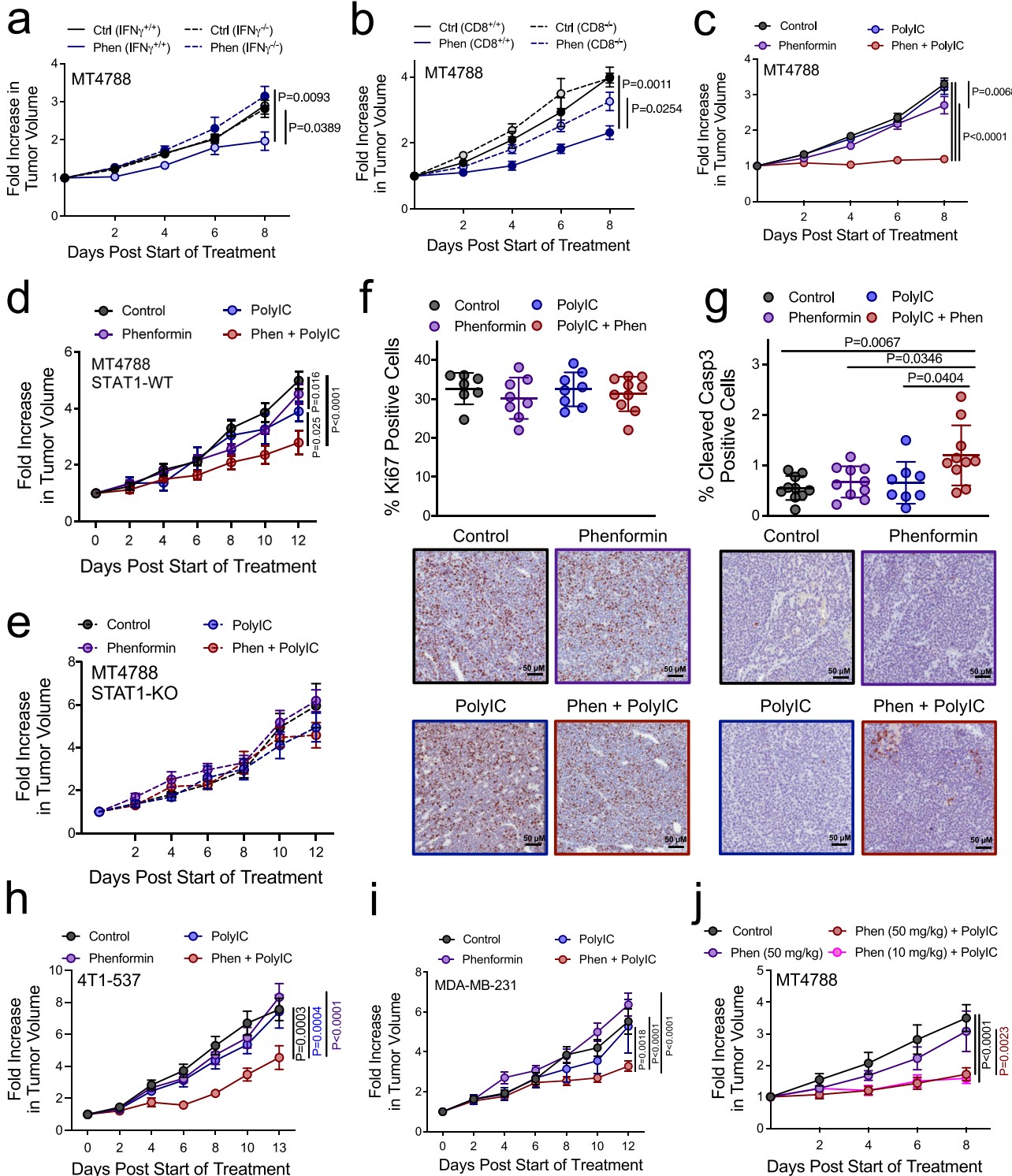

the ability of this combination treatment to relieve immune suppression.

We next tested whether therapies that stimulate STAT1-mediated inflammatory responses may increase the therapeutic efficacy of phenformin in breast cancer. We employed a well-characterized synthetic double-stranded RNA analog, polyinosinic–polycytidylic acid (polyIC), which is a toll-like receptor 3 and retinoic acid-inducible gene I receptor agonist that induces IFN-driven STAT1 activation[31]. Whereas polyIC treatment had no impact on MT4788 tumor growth, coadministrating polyIC with phenformin impaired the growth potential of these

tumors (Fig. 2c). Moreover, polyIC enhanced the tumoricidal properties of phenformin against STAT1-wild-type (WT) (Fig. 2d), but not STAT1-deficient (Fig. 2e), tumors. This suggests that the ability of polyIC to sensitize breast cancers to the tumoricidal effects of phenformin requires an intact STAT1 pathway. Immunohistochemical (IHC) analysis confirmed increased STAT1 levels in polyIC-treated tumors (Supplementary Figure 2c), whereas phenformin-treated tumors showed elevated phospho-AMPK levels (Supplementary Figure 2d). We observed no significant changes in Ki67 staining between control and treatment groups (Fig. 2f). However, polyIC and phenformin, in

**Fig. 2 PolyIC-induced STAT1 activation sensitizes breast tumors to the tumoricidal effects of phenformin in vivo. a, b** Mammary fat pad injection (MFP) of MT4788 cells into immunocompetent (FVB) mice or **a** IFNγ$^{-/-}$ or **b** CD8$^{-/-}$ animals. At ~150 mm$^3$, mice were treated with PBS or phenformin (50 mg/kg daily). **a** Control: $n = 7$; phenformin: $n = 8$ tumors/group. **b** Control CD8$^{+/+}$: $n = 5$; phenformin CD8$^{+/+}$: $n = 8$; control CD8$^{-/-}$: $n = 12$; phenformin CD8$^{-/-}$: $n = 11$ tumors. **c** MFP injection of MT4788 cells into FVB mice. At ~100 mm$^3$, mice were treated with polyIC (50 µg, daily) or saline. Two days later, when tumors were ~150 mm$^3$, phenformin (50 mg/kg, daily) (or PBS) treatment was started, in combination with polyIC or saline (every 2 days). Control: $n = 18$; phenformin: $n = 20$; polyIC: $n = 17$; phenformin + polyIC: $n = 17$ tumors. **d, e** MFP injection of **d** MT4788-STAT1-WT or **e** STAT1-KO cells into FVB mice. At ~80 mm$^3$ tumor volume, mice were treated with polyIC or saline. Two days later, when tumors were ~120 mm$^3$, phenformin (50 mg/kg, daily) (or PBS) treatment was started, in combination with polyIC or saline (every 2 days). Control: **d** $n = 11$ and **e** $n = 10$; phenformin (**d, e**): $n = 7$; polyIC: **d** $n = 6$ and **e** $n = 7$; phenformin and polyIC: **d** $n = 8$ and **e** $n = 9$ tumors. **f, g** Immunohistochemical staining of tumors described in (**c**) using **f** Ki67 and **g** cleaved caspase-3-specific antibodies. The data are shown as mean % positive cells ± SEM and is representative of **f** control: $n = 7$; polyIC: $n = 8$; phenformin: $n = 8$; phenformin + polyIC: $n = 10$ (**g**) $n = 10$ except polyIC: $n = 8$ tumors. Representative images are shown (scale bar = 50 µm). **h, i** MFP injection of **h** 4T1-537 cells into Balb/c mice and **i** MDA-MB-231 cells into SCID-Beige mice. At ~150 mm$^3$, mice were treated as described in (**c**). **h** Control: $n = 8$; phenformin, polyIC: $n = 9$; phenformin + polyIC: $n = 10$ tumors; **i** control: $n = 12$; phenformin: $n = 10$; polyIC: $n = 9$; phenformin + polyIC: $n = 13$ tumors/group. **j** MFP injection of MT4788 breast cancer cells into FVB mice. Mice were treated as described in (**c**) using two concentrations of phenformin (10 or 50 mg/kg). $n = 8$ tumors/group except phenformin (10 mg/kg) + polyIC: $n = 9$. For **a–e** and **h–j**, data are represented as the mean fold change in tumor volume relative to the start of treatment ± SEM. P values are in the Figure and were calculated using a two-way ANOVA with a Tukey's post hoc test or one-way ANOVA using Tukey's post hoc test (**f, g**). See also Supplementary Figure 2.

combination, significantly increased cleaved caspase-3 levels relative to tumors treated with either drug individually or PBS control (Fig. 2g). Combined, these observations suggest that polyIC treatment cooperates with phenformin by inducing STAT1-dependent breast cancer cell apoptosis.

We next sought to determine the generalizability of these observations by measuring the tumoricidal properties of combined polyIC/phenformin treatment using independent preclinical models of murine and human breast cancer. These include 4T1-537 cells, a lung metastatic variant that is syngeneic in immunocompetent Balb/c mice as well as in a human model of triple-negative breast cancer (MDA-MB-231) that forms tumors in immunodeficient (SCID-Beige) animals. Remarkably, polyIC/phenformin combination treatment also impaired 4T1-537 and MDA-MB-231 tumor growth in contrast to each drug as a monotherapy, which minimally impacted disease progression (Fig. 2h, i). The combinatorial effect of polyIC/phenformin treatment against MDA-MB-231 tumors further reinforces the fact that an adaptive immune response does not contribute significantly to this phenotype (Fig. 2i). Rather, innate inflammatory responses likely underpin increased sensitivity to this drug combination. To address the potential impact of our findings, we asked whether polyIC treatment could sensitize tumors to lower doses of phenformin that would be more readily achievable and associated with reduced toxicity in humans[6,15]. Coadministration of polyIC elicited comparable antitumorigenic responses using a five-fold decrease in the dose of phenformin (10 vs 50 mg/kg) (Fig. 2j). Collectively, these data support the hypothesis that polyIC-driven inflammation sensitizes tumors to the tumoricidal effects of more clinically relevant phenformin concentrations in multiple preclinical models of breast cancer.

**IFNγ minimally impacts phenformin-induced energetic stress in breast cancer cells.** We show that polyIC neither increased phospho-AMPK levels nor induced infiltration of granzyme B-positive cells into phenformin-treated tumors (Supplementary Figure 2d, e). These data support the concept that the mechanisms of cooperativity observed with this drug combination extend beyond amplification of energetic stress or induction of antitumor immune responses. However, given the established role of phenformin in inhibiting mitochondrial metabolism[3–5], we assessed whether energetic stress underlies IFNγ-induced sensitivity of tumors to this biguanide. Phenformin monotherapy obliterates the oxygen consumption rate (OCR) in breast cancer cells (Fig. 3a), including dramatically reduced rates of coupled and uncoupled respiration (Fig. 3b–f) and a profound reduction

in OCR-coupled ATP production (Fig. 3g). IFNγ treatment modestly reduced the basal and maximal respiration rates of breast cancer cells (Fig. 3b, c) and in a STAT1-dependent manner (Supplementary Figure 3a, b). Despite this fact, IFNγ did not further potentiate phenformin-induced inhibition of cellular respiration (Fig. 3b, c). Consistently, IFNγ-driven STAT1 activation led to a 30% reduction in the bioenergetic capacity of breast cancer cells compared to the 25-fold reduction observed with phenformin treatment, either alone or combined with IFNγ (Fig. 3h and Supplementary Figure 3c). These results suggest that IFNγ-driven STAT1 activation in breast cancer cells does not further potentiate the profound inhibition of cellular respiration induced by phenformin.

We next assessed whether IFNγ altered the metabolic flexibility of breast cancer cells in response to phenformin treatment. As expected, phenformin increased the extracellular acidification rate (ECAR) in breast cancer cells, yet no further differences were measured upon co-treatment with IFNγ (Fig. 3i). Indeed, similar ECAR values were observed in breast cancer cells irrespective of whether they were treated with IFNγ or retained an intact STAT1 pathway (Supplementary Figure 3d). Moreover, IFNγ modestly decreased the metabolic flexibility of MT4788 breast cancer cells in a STAT1-dependent manner (Fig. 3j, k and Supplementary Figure 3e, f). However, phenformin, alone or when combined with IFNγ, obliterates ATP production from oxidative phosphorylation (OXPHOS) with comparable increases in energy derived from glycolysis (Fig. 3j, k and Supplementary Figure 3e, f). Previous studies suggest that STAT1 activation may increase glycolysis[32,33]. Indeed, increased pyruvate and lactate levels, along with an elevated lactate/pyruvate ratio, were observed in IFNγ-stimulated STAT1-wild-type and not STAT1-deficient cells (Supplementary Figure 3g, h). Co-treatment with IFNγ and phenformin modestly increased steady-state pyruvate and lactate steady levels, although there were no differences in the lactate/pyruvate ratio compared to phenformin treatment alone (Fig. 3l, m). IFNγ treatment had no impact on breast cancer cell viability following glucose deprivation (Supplementary Figure 3i), suggesting that IFNγ signaling does not alter the glucose dependency of breast cancer cells.

Moreover, IFNγ stimulation enhances phenformin-induced α-ketoglutarate (αKG) levels as well as the αKG/citrate ratio in breast cancer cells (Fig. 3n, o). Finally, this IFNγ-induced increase in αKG levels and the αKG/citrate ratio is STAT1-dependent (Supplementary Figure 3j, k). These data demonstrate increased reductive carboxylation of glutamine metabolism, as previously reported with electron transport chain inhibitors[34,35]. In support

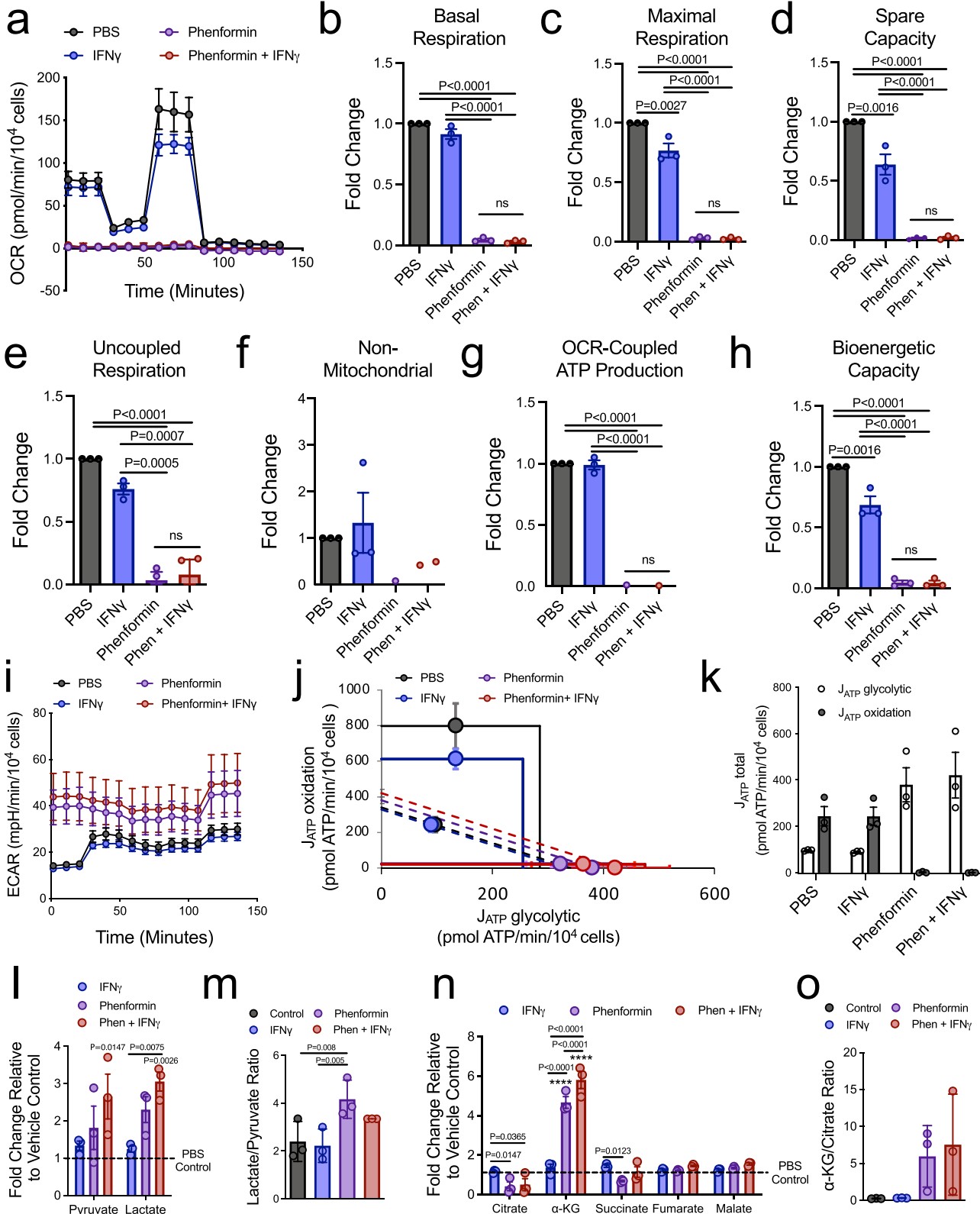

of this, IFNγ treatment leads to a modest but statistically significant increase in breast cancer viability in response to reduced glutamine levels (Supplementary Figure 3l). While interesting, these observations cannot explain the increased synergy observed with IFNγ and phenformin in stimulating cytotoxic responses in breast cancer cells. These results suggest that although IFNγ-driven STAT1 activation modestly reduces

the bioenergetic capacity and flexibility of breast cancer cells, these differences pale in comparison to the greater ability of phenformin to potentiate energetic and biosynthetic stress.

**IFNγ and polyIC-induced phenformin sensitivity relies on mitochondrial ROS production in breast cancer cells.** By inhibiting complex I of the electron transport chain, phenformin

**Fig. 3 IFNγ induces moderate energetic stress in breast cancer cells. a** Oxygen consumption rate (OCR), $n = 3$ independent experiments (mean of means) ± SEM. **b–f** Fold change in the rates of **b** basal respiration, **c** maximal respiration, **d** spare capacity, **e** uncoupled respiration, and **f** non-mitochondrial respiration from the samples analyzed in (**a**). $n = 3$ independent experiments, (mean of means) ± SEM. **g, h** Fold change in **g** OCR-coupled ATP production and **h** the bioenergetic capacity of cells described in (**a**). The data are presented as a mean of means ± SEM, of $n = 3$ independent experiments. **i** Extracellular acidification rate (ECAR) from cells described in (**a**). The data are representative of $n = 3$ independent experiments (mean of means) ± SEM. **j** The metabolic capacity and flexibility of cells were represented by plotting the basal (point on the dotted line) and maximal rates (point on solid line) of ATP production from glycolysis ($J_{ATP}$ glycolytic) and oxidative phosphorylation ($J_{ATP}$ oxidation), upon treatment. $n = 3$ independent experiments, (mean of means) ± SEM. **k** Total basal ATP production from either glycolysis or oxidation, upon treatment. $n = 3$ independent experiments, presented as mean of means ± SEM. **l** Fold change in steady-state levels of glycolytic metabolites. The data are representative of $n = 3$ independent experiments (mean of means) ± SEM. **m** The lactate/pyruvate ratio was determined from the samples analyzed in (**l**), of $n = 3$ independent experiments (mean of means) ± SD. **n** Fold change in steady-state levels of citric acid cycle metabolites, of the same samples as (**l,**) $n = 3$ independent experiments (mean of means) ± SEM. ****$P < 0.0001$ compared to PBS control. Other $P$ values indicated in the figure. **o** α-Ketoglutarate/citrate ratio was determined from (**n**), of $n = 3$ independent experiments (mean of means) ± SD. For each panel, MT4788 cells were treated with 1 ng/mL IFNγ, phenformin 500 μM, combination, or PBS treatment as the vehicle control. $P$ values were calculated using a two-way ANOVA with a Tukey's post hoc test and are indicated in the figure or above. See also Supplementary Figure 3. n.s. not significant.

---

impedes cellular respiration leading to the production of mitochondrial ROS[4,24]. Mitochondrial superoxide anion production is increased in MT4788 and MDA-MB-231 cells following phenformin treatment (Fig. 4a, b). This was corroborated using a probe that measures total ROS levels (Supplementary Figure 4a–c). IFNγ treatment neither increased ROS levels in breast cancer cells nor potentiated phenformin-induced ROS production (Fig. 4a, b and Supplementary Figure 4a–c). Thus, IFNγ signaling is not sufficient to induce oxidative stress. Tumor cells must cope with elevated ROS levels compared to their non-transformed counterparts and exploit moderately raised ROS levels to promote tumor growth and metastasis. Along this line, numerous studies have explored the therapeutic potential of exploiting ROS-induced oxidative damage to selectively kill cancer cells[25]. We examined whether IFNγ stimulation sensitized breast tumors to the cytotoxic effects of phenformin-induced ROS production. We show that MitoTEMPO, a mitochondrial ROS scavenger, reversed the cytotoxic effects of IFNγ/phenformin combination treatment in three independent breast cancer models (MT4788, MDA-MB-231, and BT474) (Fig. 4c–e). Moreover, co-injection of mice with MitoTEMPO restored the growth potential of mammary tumors treated with polyIC/phenformin combination therapy (Fig. 4f). Consistent with these observations, 8-oxo-2′-deoxyguanosine (8-oxo-dG) IHC staining showed an ~1.6-fold increase in oxidative DNA damage in polyIC/phenformin-treated tumors compared to either the phenformin-treated group or control tumors (Fig. 4g). Thus, oxidative stress underlies the synergistic effects of phenformin and IFNγ-driven STAT1 activation in potentiating breast cancer cell death. Combined with the observation that IFNγ stimulation does not increase ROS levels, these data further support the hypothesis that IFNγ likely perturbs the ROS-scavenging potential of breast cancer cells.

**Inhibiting glutathione synthesis sensitizes breast cancer cells to phenformin.** Glutathione is a non-protein thiol-containing molecule and major ROS scavenger in cancer cells. It exists in thiol reduced (GSH) or oxidized (GSSG) states, whereby GSH predominates. NADPH is an essential cofactor and electron donor to replenish GSH levels and maintain redox balance[36]. We first assessed whether IFNγ or phenformin, alone and in combination, altered glutathione levels in breast cancer cells. Phenformin treatment reduced GSH levels and decreased the GSH/GSSG ratio in MT4788 cells (Supplementary Figure 4d), which is consistent with its ROS-inducing properties (Fig. 4a, b and Supplementary Figure 4a–c). However, IFNγ treatment, alone or combined with phenformin, did not further impact glutathione levels or the GSH/GSSG ratio (Supplementary Figure 4d).

Moreover, similar trends in the regulation of glutathione levels in STAT1-deficient cells (Supplementary Figure 4e). Finally, phenformin increased the NADH/NAD+ ratio in breast cancer cells, which is reflective of its role as a complex I inhibitor (Supplementary Figure 4f). Although AMPK activation has been shown to reduce NAPDH consumption[37], IFNγ and/or phenformin treatment had no impact on the NADPH/NADP+ ratio in MT4788 breast cancer cells (Supplementary Figure 4g). Moreover, steady-state levels of amino acid constituents of glutathione (Glu, Cys, and Gly) are unaffected by IFNγ in MT4788 breast cancer cells (Supplementary Figure 4h). These results suggest that IFNγ-driven STAT1 activation does not directly impair glutathione production.

However, we were intrigued by the ability of phenformin to reduce the glutathione-buffering capacity of breast cancer cells. Therefore, we examined whether further perturbation of glutathione production in breast cancer cells could sensitize them to the antitumorigenic effects of phenformin. Indeed, pharmacological inhibitors of glutathione synthesis sensitize tumors to ROS-inducing chemotherapies[38]. We, therefore, tested whether buthionine sulfoximine (BSO), which inhibits the first step in glutathione synthesis, could potentiate the cytotoxic effects of phenformin in breast cancer. Whereas BSO treatment had no impact on cell viability, it increased the sensitivity of MDA-MB-231 (4.6-fold) and BT474 (5.7-fold) cells to phenformin treatment (Fig. 5a and Supplementary Figure 5a). Moreover, BSO co-treatment elicited comparable antitumorigenic effects in combination with a 5-fold (100 μM) and 25-fold (20 μM) reduction in phenformin levels in MDA-MB-231 and BT474 cells, respectively (Fig. 5b and Supplementary Figure 5b). Finally, MitoTEMPO rescued the viability of cancer cells treated with BSO and phenformin (Fig. 5c). These results suggest that glutathione synthesis inhibitors profoundly sensitize breast cancer cells to phenformin by inducing oxidative stress. Although BSO and phenformin co-treatment induced a modest reduction in the number of bromodeoxyuridine-positive cells (1.3-fold), we observed a more robust increase in the frequency of Annexin V/propidium iodide-positive (PI+) cells (2.7-fold) in response to this drug combination (Fig. 5d–e and Supplementary Figure 5c). These data suggest that the tumoricidal effects of combined BSO and phenformin treatment are predominately a result of increased apoptosis.

Phenformin does not require OCT transporters to enter the cell, making it a more potent complex I inhibitor at lower concentrations[4,39]. However, most studies in oncology have focused on the related family member, metformin[6]. This is attributed to the approved use of metformin in the long-term management of type 2 diabetes mellitus[40], owing to its lower rates

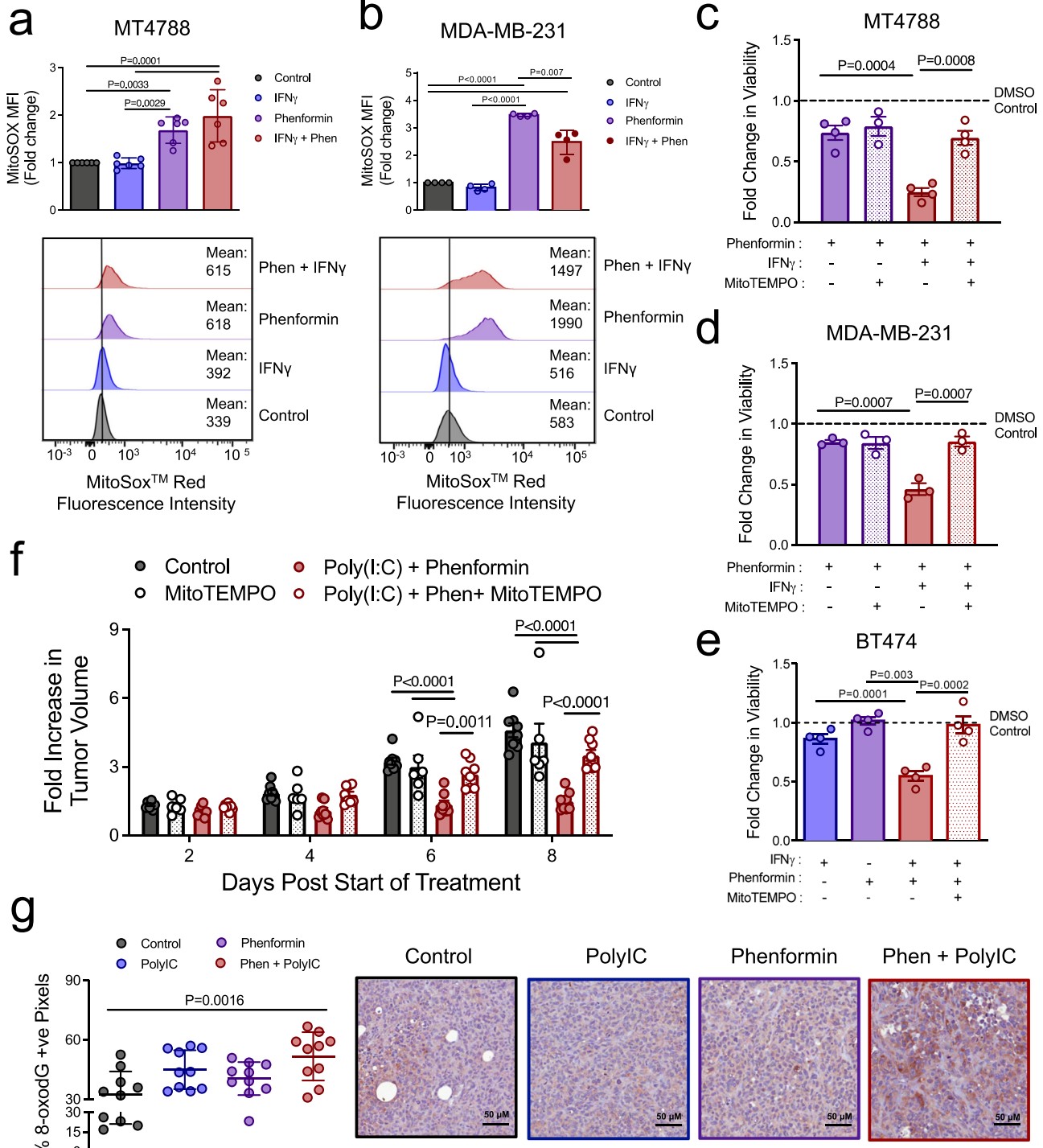

**Fig. 4 IFNγ and polyIC-induced phenformin sensitivity requires mitochondrial ROS. a**, **b** MitoSOX geometric mean fluorescence intensity (MFI) and representative histograms of **a** MT4788 and **b** MDA-MB-231 cells treated with IFNγ and phenformin, alone or in combination, for 24 h. The data are shown as the fold change in MFI compared to PBS controls ± SEM: **a** MT4788: $n = 6$/group; **b** MDA-MB-231: $n = 4$/group. See Supplementary Figure 10 for gating strategy. **c–e** Fold change in cell viability compared to DMSO control of **c** MT4788, **d** MDA-MB-231, and **e** BT474 cells treated for 48 h with IFNγ and/or phenformin, either in the absence or presence of 10 μM MitoTEMPO. Data are presented as the mean of means ± SEM, of $n = 3$ (**d**) or $n = 4$ (**c**, **e**) independent experiments. **f** Mammary fat pad injection of MT4788 breast cancer cells into FVB mice. At ~100 mm³, mice were treated with vehicle control or polyIC (50 μg every 2 days), 2 days later phenformin (50 mg/kg, daily) (or PBS) treatment was initiated with or without 3 mg/kg MitoTEMPO. Data are represented as a mean fold increase in tumor volume relative to the start of combination treatment ± SEM. Control group: $n = 8$; MitoTempo: $n = 6$; phenformin + polyIC: $n = 8$; phenformin + polyIC + MitoTempo: $n = 8$ tumors. **g** 8-oxo-dG immunohistochemical staining of paraffin-embedded MDA-MB-231 tumors as described in (Fig. 2g). The data are represented as percent positive pixels mean ± SD ($n = 10$ independent tumors/group). Representative images are also shown. *P* values were calculated using a two-way ANOVA with a Tukey's post hoc test and are shown in the figure. See also Supplementary Figure 4.

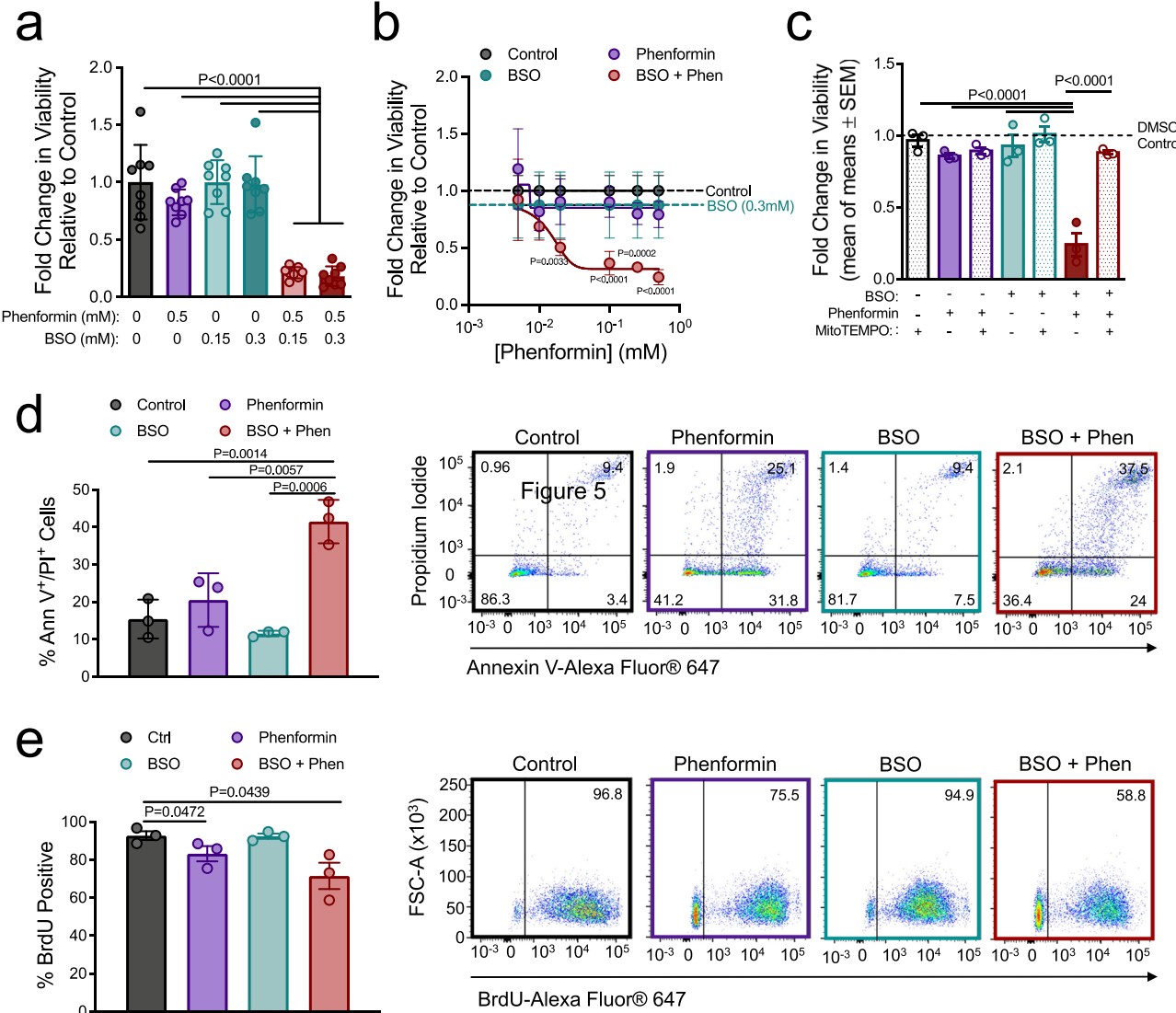

**Fig. 5 Inhibiting glutathione synthesis sensitizes breast cancer cells to phenformin. a–c** MDA-MB-231 cells were treated for 24 h with varying concentrations of **a** BSO or **b** phenformin as indicated. **c** Phenformin/BSO-treated cells were also pretreated with 10 μM MitoTEMPO. The data are shown as fold change in viability compared to PBS control. For (**a**, **b**), the data are representative of duplicate experiments (n = 4 technical repeats) mean ± SD. For (**c**), the data are shown from n = 3 independent experiments, (mean of means) ± SEM. P values indicated in (**b**) compare the combination of phenformin + BSO to treatment with the respective concentration of phenformin alone. **d**, **e** Percentage of **d** Annexin V+/PI+ or **e** BrdU-positive MDA-MB-231 cells (±SEM) as determined by flow cytometry. Cells were treated with and PBS control, phenformin, and/or BSO for 40 h. For each panel, the data are representative of three independent experiments. Representative dot plots are shown. See Supplementary Figure 10 for gating strategy. P values are indicated in the figure and were calculated using a two-way ANOVA with a Tukey's post hoc test. See also Supplementary Figure 5.

of lactic acidosis[41]. We, therefore, assessed the relative ability of BSO to sensitize breast cancer cells to the cytotoxic effects of metformin vs phenformin. Whereas low doses of phenformin (100 μM) synergized with BSO to reduce cancer cell viability, metformin could not potentiate BSO-induced cell death, even at 10-fold higher concentrations (1 mM) (Supplementary Figure 5d). Unlike phenformin, metformin was unable to stimulate ROS production at these concentrations (Supplementary Figure 5e). These data show that pharmacological inhibitors of glutathione synthesis selectively and potently sensitize breast cancer cells to phenformin by inducing oxidative stress and subsequent apoptotic cell death.

**Inhibition of NQO1 levels potentiates the ROS-dependent tumoricidal effects of phenformin.** These studies demonstrate the clinical potential of combining BSO and phenformin to treat

individuals with breast cancer. However, they do not inform how IFNγ-driven STAT1 activation potentiates the ROS-dependent, cytotoxic effects of this biguanide. To address this, we performed genome-wide RNA-sequencing (RNA-seq) analysis on control and STAT1-deficient breast cancer cells (MT864 and MT4788) following IFNγ stimulation. We identified 1233 genes that were differentially expressed between MT4788-STAT1-WT vs MT4788-STAT1-knockout (KO) cells and 573 genes differentially expressed between MT864-STAT1-WT vs MT864-STAT1-KO cells (Supplementary Data File 1 and Supplementary Data File 2; >100 reads, >2-fold, false discovery rate < 0.05). GO term analysis showed that transcriptional responses related to immune system processes, anti-viral responses, and antigen processing and pre-sentation were most strongly upregulated by IFNγ-driven STAT1 activation (Supplementary Data File 3 and Supplementary Figure 6a, b).

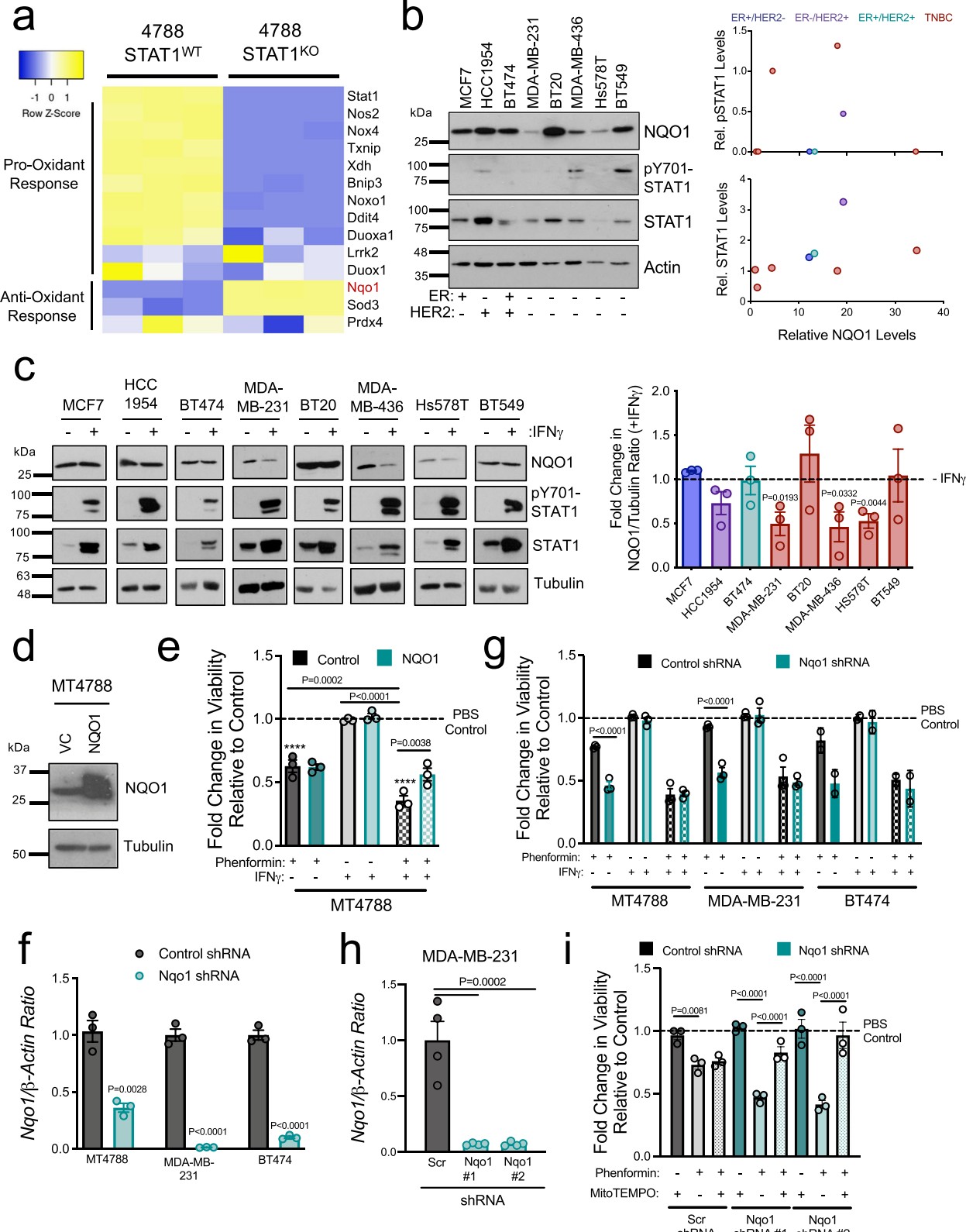

With the knowledge that oxidative stress is required for IFNγ and phenformin to elicit antitumorigenic effects (Fig. 4c–f), we focused on differentially expressed genes that control redox homeostasis (Supplementary Figure 6b). This included ten genes (*Nos2*, *Nox4*, *Txnip*, *Xdh*, *Bnip3*, *Noxo1*, *Ddit4*, *Duoxa1*, *Lrrk2*, *Duox1*), which encode a protein with ROS-inducing properties that were overexpressed in STAT1-WT compared to STAT1-

deficient cells (Fig. 6a and Supplementary Figure 6c). However, as IFNγ does not stimulate ROS production in breast cancer cells (Fig. 4a, b and Supplementary Figure 4a–c), we focused on differentially expressed genes that function as ROS scavengers. Three such genes were identified, including *PRDX4*, *SOD3*, and *NQO1* (Fig. 6a and Supplementary Figure 6c). *Prdx4* messenger RNA (mRNA) levels were increased by IFNγ and *Sod3* mRNA

**Fig. 6 IFNγ-induced inhibition of NQO1 expression potentiates the antitumorigenic effects of phenformin. a** RNA-seq analysis of MT4788-VC and STAT1-KO cells stimulated with IFNγ for 24 h. Heatmap of differentially expressed genes (>2-fold; FDR < 0.05) associated with ROS metabolism. **b** Immunoblot analysis of human breast cancer cell lines. Relative NQO1 protein levels compared to pY701-STAT1 or total STAT1 levels were quantified, $n = 1$ technical repeat. **c** Immunoblot analysis of cell lines from (**b**) following 48 h IFNγ treatment. Fold change of the NQO1/tubulin ratio upon IFNγ treatment relative to PBS controls was quantified from $n = 3$ independent experiments, (mean of means) ± SEM. **d** Immunoblot analysis of vector control (VC) and NQO1-overexpressing MT4788 cells, representative of $n = 3$ biological repeats. **e** Relative viability of cells described in (**d**) in response to phenformin (500 μM) and IFNγ treatment (48 h). Data are shown as fold change in viability compared to PBS-treated controls and is representative of $n = 3$ independent experiments (mean of means) ± SEM. ****$P$ value < 0.0001 comparing with PBS control. **f** RT-qPCR analysis of cell lines transduced with shRNAs targeting human or mouse *NQO1* or with a control non-mammalian shRNA. Data are presented as mean of means ± SEM, of $n = 3$ biological repeats. **g** Cells in (**f**) were tested for their relative sensitivity to IFNγ and/or phenformin (48 h). The data are shown as fold change in cell viability compared to PBS control and is representative of $n = 3$ independent experiments (MT4788 and MDA-MB-231) (mean of means) ± SEM, or two independent experiments (BT474) (mean of means) ± SD. **h** RT-qPCR analysis of MDA-MB-231 cells engineered to individually express two shRNAs targeting human *NQO1* or a control with non-mammalian targeting shRNA. $n = 4$ biological repeats; (mean of means) ± SEM. **i** Cells described in (**h**) were tested for relative sensitivity to phenformin (48 h), either in the absence or presence of 5 μM MitoTEMPO. Data are shown as fold change in viability compared to PBS controls, $n = 3$ independent experiments (mean of means) ± SEM. $P$ values were calculated using unpaired two-sided $t$ tests comparing IFNγ and PBS treatment (**c, f**), a two-way ANOVA with a Tukey's post hoc test (**e, g, h, i**) and a one-way ANOVA with a Tukey's post hoc test (**h**). See also Supplementary Figures 6 and 7.

levels were only repressed by IFNγ in MT4788 cells (Supplementary Figure 6c). In contrast, STAT1 activation reduced *Nqo1* levels in both MT864 and MT4788 cells in response to IFNγ stimulation (Supplementary Figure 6c). NQO1 encodes an NAD(P)H dehydrogenase that functions as a two-electron reductase with important roles in superoxide scavenging, quinone detoxification, and the cellular stress response[42]. Moreover, NQO1 is frequently overexpressed in many tumor types, including lung and breast cancers[43,44]. We validated *Nqo1* as a STAT1 target gene in MT4788 cells whose expression decreases upon IFNγ treatment, specifically in STAT1-proficient cells (Supplementary Figure 7a, b). We next compared steady-state NQO1, STAT1, and pY701-STAT1 protein levels across a panel of human breast cancer cell lines spanning various subtypes. Baseline STAT1 and pY701-STAT1 levels did not correlate with differences in NQO1 expression levels (Fig. 6b). However, we observed decreased NQO1 protein levels upon IFNγ treatment in three of five triple-negative breast cancer (TNBC) cell lines tested (MDA-MB-231, Hs578T, and MDA-MB-436). In contrast, IFNγ stimulation did not appreciably alter NQO1 levels in any of the ER$^+$ or HER2$^+$ cell lines tested (Fig. 6c). Quantitative reverse transcription-PCR (RT-qPCR) analysis did not show a similar decrease in *NQO1* levels in IFNγ-treated TNBC cells, suggesting that IFNγ stimulation predominately reduces NQO1 levels in human breast cancer cells at the post-transcriptional level (Supplementary Figure 7c). Immunoblot analysis of tumor lysates from 18 patient-derived xenografts (PDXs) of lung cancer brain metastases confirmed this inverse correlation (Supplementary Figure 7d)[45]. These results suggest that *NQO1* is a STAT1-regulated gene that is repressed in response to IFNγ stimulation in some but not all breast cancers. Our data further suggest complex mechanisms by which STAT1 controls NQO1 expression, including transcriptional and post-transcriptional control, in different biological contexts.

To address whether IFNγ-induced inhibition of NQO1 expression contributes to the observed cooperation between IFNγ and phenformin, we overexpressed NQO1 in MT4788 cells (Fig. 6d). Increasing NQO1 levels rescued their viability in response to combined IFNγ and phenformin treatment (Fig. 6e). We also employed mouse and human short hairpin RNAs (shRNAs) to silence *NQO1* expression levels in multiple breast cancer cell lines (Fig. 6f). Reducing NQO1 expression impaired the viability of MT4788, BT474, and MDA-MB-231 cells in response to phenformin, which approximated the reduction in viability observed with IFNγ/phenformin combination treatment (Fig. 6g). Breast cancer cells expressing NQO1 shRNAs are not

further sensitized to IFNγ and phenformin co-treatment, suggesting that *NQO1* is an essential target gene that confers the antitumorigenic effects of phenformin (Fig. 6g). Finally, we examined whether IFNγ-mediated inhibition of NQO1 expression sensitizes tumors to phenformin through an oxidative stress response. While shRNA-mediated *NQO1* knockdown increased the cytotoxic potential of phenformin in MDA-MB-231 cells, these antitumorigenic effects were reversed by co-treatment with MitoTEMPO (Fig. 6h, i). These findings are in accordance with previous studies showing that ROS-induced cytotoxicity of rotenone, another mitochondrial complex I inhibitor, could be reversed with coenzyme Q (CoQ) and in an NQO1-dependent manner[46]. Collectively, these results demonstrate that STAT1-dependent signaling inhibits NQO1 expression in some breast cancer cells, sensitizing them to phenformin-induced oxidative stress. Direct silencing of NQO1 expression reveals an important role for NQO1 in protecting tumor cells from phenformin-generated oxidative stress.

**β-Lapachone, an NQO1-bioactivatable drug, sensitizes breast tumors to phenformin by inducing oxidative damage.** NQO1 is a viable drug target in oncology[42]. β-Lapachone is a quinone-containing prodrug that is bioactivated by NQO1 to undergo a futile redox cycle. In doing so, β-lapachone not only sequesters NQO1 from its endogenous substrates but also further potentiates superoxide generation in NQO1-positive cells[22,47,48]. We examined whether combined β-lapachone and phenformin treatment could also elicit antitumorigenic responses. While β-lapachone minimally impacted cell viability, it profoundly sensitized MDA-MB-231, BT474, and BT549 cells to phenformin treatment (Fig. 7a and Supplementary Figure 8a, b). Whereas β-lapachone/phenformin treatment did not appreciably induce apoptosis (Fig. 7b), this drug combination significantly decreased tumor cell proliferation (Fig. 7c). We extended these findings in vivo and showed that combined β-lapachone and phenformin treatment significantly impaired MDA-MB-231 mammary tumor growth (Fig. 7d). Although we do not observe steady-state differences in the percentage of Ki67 and cleaved caspase-3-positive cells at the experimental endpoint (Supplementary Figure 8c, d), combined treatment with phenformin and β-lapachone significantly increased oxidative damage in mammary tumors as assessed by 8-oxo-dG IHC staining (32.7% in control tumors vs 49.2% in β-lapachone/phenformin-treated tumors) (Fig. 7e). These data demonstrate that β-lapachone sensitizes breast tumors to the antineoplastic effects of phenformin by inducing oxidative DNA damage.

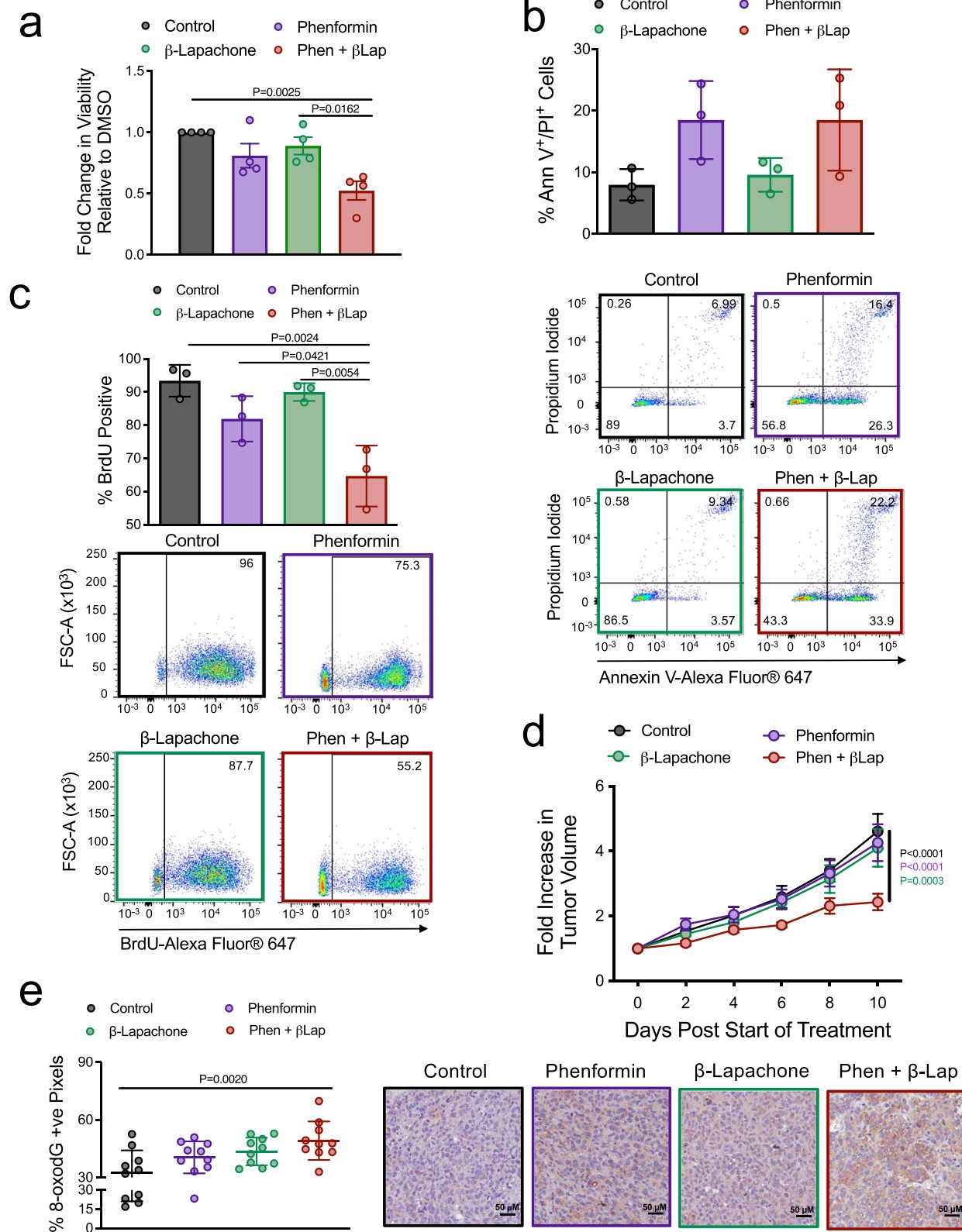

**Inhibiting targetable ROS-scavenging mechanisms selectively sensitizes human breast cancers to multiple mitochondrial complex I inhibitors.** Tumors must cope with chronically elevated ROS levels compared to normal cells, exposing a selective vulnerability for cancer therapies that tip the balance to favor increased ROS generation or decreased ROS scavenging[49]. To test whether phenformin-induced ROS levels selectively inhibit the viability of transformed cells, we employed immortalized NMuMG cells and an isogenic cell line that was transformed with NeuNT, an oncogenic variant of ErbB2 (NMuMG-NT)[50]. As expected, NMuMG cells generated lower ROS levels following phenformin treatment compared to their NeuNT-transformed counterparts (Fig. 8a). Consistently, NMuMG cells were more resistant to the cytotoxic effects of phenformin when combined

**Fig. 7 β-Lapachone, an NQO1-bioactivatable drug, synergistically sensitizes breast tumors to phenformin by inducing oxidative damage. a** Viability of MDA-MB-231 cells treated with phenformin and/or β-lapachone for 48 h. Data are shown as fold change in viability relative to DMSO control and is representative of $n = 4$ independent experiments (mean of means) ± SEM. **b, c** Percentage of **b** Annexin $V^+$/PI$^+$ or **c** BrdU-positive MDA-MB-231 cells (mean of means ± SEM) as determined by flow cytometry. Cells were treated with PBS control, phenformin, and/or β-lapachone for 48 h. The data are representative of three independent experiments. Representative dot plots are shown. **d** Mammary fat pad injection of MDA-MB-231 breast cancer cells into SCID-Beige mice. At tumor size ~100 mm$^3$, mice were started on β-lapachone/HPβCD (25 mg/kg, every 2 days) or (HPβCD/PBS). Two days later, phenformin (50 mg/kg, daily) (or PBS) was initiated, in combination with vehicle (HPβCD/PBS) or β-lapachone/HPβCD. Data are represented as fold change in tumor volume relative to the start of combination treatment ± SEM, $n = 11$ tumors/group; except β-lapachone/HPβCD, $n = 12$ tumors/group. $P$ values indicated in the figure comparing combination treatment group to: black font: control; purple font: phenformin; green font: β-lapachone groups. **e** 8-oxo-dG immunohistochemical staining of paraffin-embedded tumors as described in (**d**). The data are represented as the mean percent positive pixels ± SEM ($n = 10$ tumors/group). Representative images are also shown. For in vitro studies, 0.5 μM β-lapachone and 500 μM phenformin were used. $P$ values are indicated in the figure and were calculated using a two-way ANOVA with a Tukey's post hoc test. See also Supplementary Figure 8.

with BSO or β-lapachone. However, NMuMG-NT cells showed significant antitumorigenic responses to both combination treatments (Fig. 8b). Thus, cancer cells are selectively vulnerable to the tumoricidal effects of phenformin in combination with drugs that potentiate oxidative stress, creating a therapeutic window of opportunity that spares non-transformed tissues.

Recent studies have described a novel small-molecule complex I inhibitor, IACS-010759, as a potent anticancer agent in leukemias that is currently in clinical trials[51]. IACS-010759 also collaborates with β-lapachone to reduce the viability of MDA-MB-231 and BT474 cells (Fig. 8c). This is consistent with the observation that β-lapachone and IACS-010759 co-treatment increases overall ROS levels (Fig. 8d). These data suggest that inhibitors that impair the ROS-scavenging potential of breast cancer cells significantly increase the therapeutic potential of several complex I inhibitors in oncology.

Finally, we assessed whether our findings could be translated to more clinically relevant models of breast cancer. We employed cell lines from six PDXs obtained from HER2$^+$ (CRC-132 and GCRC2080) and basal-like (GCRC1735, GCRC1915, GCRC1963, and GCRC1986) breast cancer[52,53]. Both HER2$^+$ PDXs showed remarkable sensitivity to the cytotoxic effects of phenformin when combined either with BSO or β-lapachone (Fig. 8e, f). Except for GCRC1963, all remaining basal-like PDXs displayed reduced viability in response to BSO/phenformin and β-lapachone/phenformin combination treatments in vitro (Fig. 8g, h). Although we do observe an inverse relationship between NQO1 and STAT1 levels in these breast cancer PDXs (Supplementary Figure 9a, b), the NQO1/STAT1 ratio is not sufficient to predict relative sensitivity to combined β-lapachone/phenformin treatment (Supplementary Figure 9c). Indeed, we observed differences in the relative ability of BSO or β-lapachone to sensitize individual PDXs to phenformin treatment (Fig. 8g, h), suggesting that breast tumors likely differ in their reliance on glutathione and/or NQO1 to maintain redox balance. Collectively, these results support the notion that targetable ROS-scavenging mechanisms can be alleviated to selectively sensitize human breast cancers to mitochondrial complex I inhibitors.

## Discussion

This study reveals a vulnerability of breast tumors to treatments combining phenformin with inhibitors that hinder antioxidant defense mechanisms (Fig. 9). This strategy exposes a selective vulnerability in cancer cells by simultaneously capitalizing on the energetic and oxidative stress induced by phenformin. Most studies in oncology have focused on metformin as this biguanide is well tolerated, has a lower risk of lactic acidosis, and is widely used for the long-term management of type 2 diabetes[6]. However, ongoing clinical trials are beginning to examine the efficacy of phenformin in melanomas (NT03026517) and compelling pre-clinical studies suggest that phenformin may be more suitable in

oncology. Unlike metformin, phenformin does not require OCT transporters for entry into the cell, is a more potent complex I inhibitor, and exerts superior antitumor effects in several cancers[54,55]. Compared to metformin, we show that phenformin is a more potent ROS inducer, even at lower concentrations. This forms the molecular basis for the selective sensitivity of breast tumors to phenformin with therapies that target tumor ROS-scavenging mechanisms. Combined, our results contribute to accumulating evidence that phenformin may be superior to metformin as a candidate biguanide for cancer therapy.

Increased mitochondrial metabolism and redox homeostasis have emerged as important factors that promote the metastatic potential of breast cancers[56,57]. Indeed, the transcriptional diversity of breast cancers relies on increased OXPHOS to promote their metastatic seeding[58]. Moreover, increased OXPHOS and ROS-scavenging mechanisms underlie low response rates to standard therapies as well as the development of residual disease[59,60]. Considering this, mitochondrial complex I inhibitors, which simultaneously block OXPHOS and potentiate ROS production, were predicted to be promising therapeutic agents for individuals with relapse and/or resistant disease[61]. Therapy-induced ROS production also contributes to the cytotoxicity of chemotherapies and ionizing radiation. Resistant cancer cells are protected by increased antioxidant defenses[62]. With these ROS-scavenging mechanisms intact, some tumors may exploit elevated ROS levels to potentiate hypoxia-inducible factor-1α signaling, leading to the development of chemo-resistant breast cancers[63]. Our data suggest that combining complex I inhibitors with inhibitors that block ROS scavengers may prevent or overcome these resistance mechanisms. Our findings set the stage for clinical trial enabling studies to select the best combinations of biguanides and inhibitors of ROS defense that are suitable as a treatment. While phenformin is an obvious candidate, novel biguanides such as IACS-010759 and IM156 also deserve consideration[51,64]. The higher drug concentrations of biguanides required to elicit the antineoplastic effects achieved in preclinical studies, have impeded our ability to translate findings to the clinic[14,15]. Indeed, pharmacokinetic studies demonstrated that individuals with diabetes treated with metformin achieve maximal plasma concentrations that are at least 6–10-fold less than what is observed in preclinical cancer models treated with metformin[14,15]. Notwithstanding the fact that similar pharmacokinetic studies have yet to be performed with phenformin or other complex I inhibitors, we showed that polyIC co-treatment elicited comparable antineoplastic effects with 5–10-fold lower doses of phenformin (10 mg/kg) than what is typically used in preclinical studies in oncology (50–100 mg/kg). Clinical trials are required to confirm these studies in individuals with cancer. Finally, ARQ-761 is a β-lapachone analog inhibitor of NQO1-mediated ROS scavenging for which phase I clinical trial data in advanced solid tumors are available[65]. Other inhibitors that

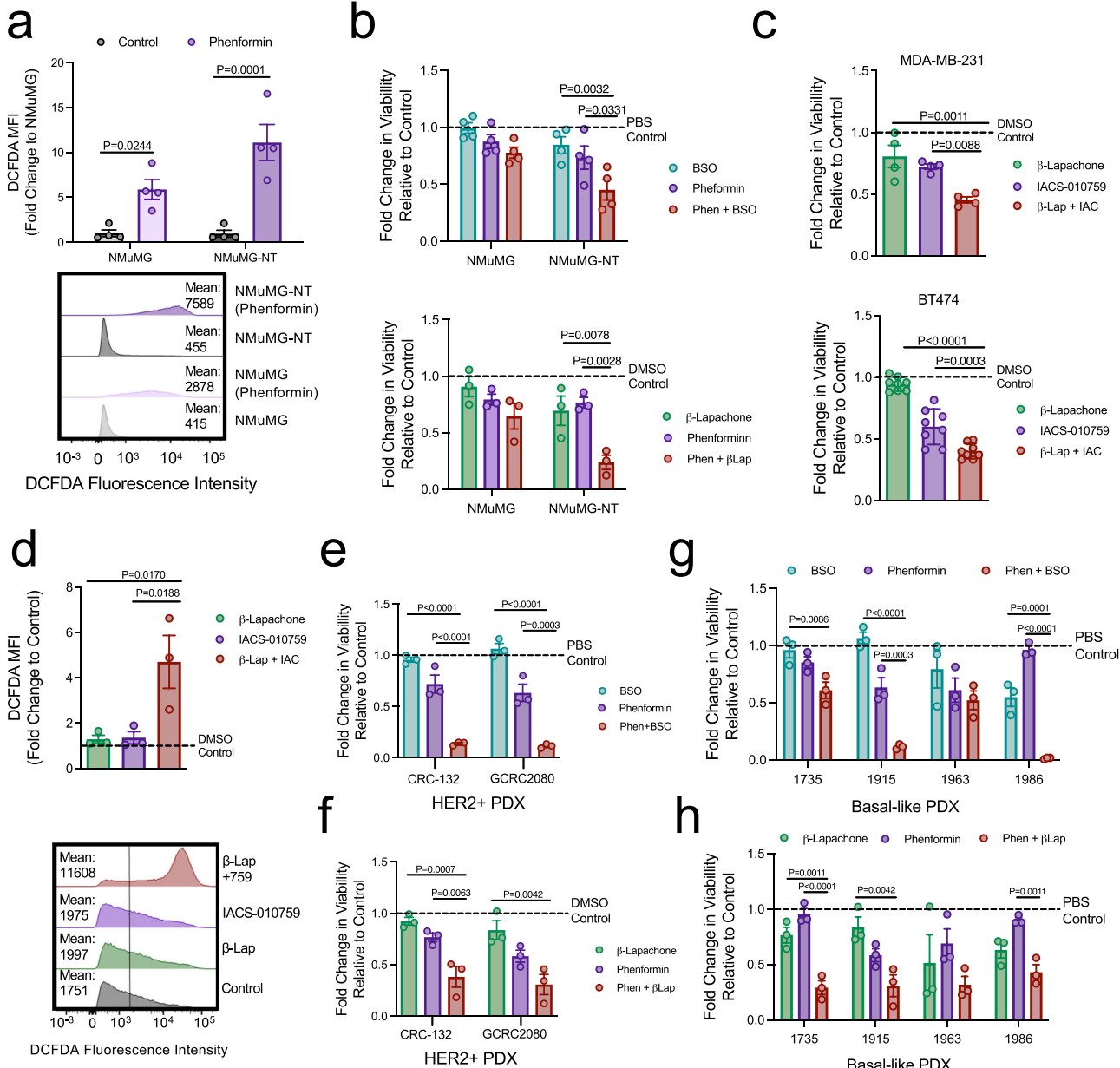

**Fig. 8 Targetable ROS-scavenging mechanisms selectively sensitize human breast cancers to multiple mitochondrial complex I inhibitors. a** DCFDA geometric mean fluorescence intensity (MFI) for immortalized NMuMG and transformed NMuMG-NeuNT cells treated with PBS or phenformin (500 μM) for 24 h. The data are shown as fold change in MFI compared to NMuMG cells and represent the mean of n = 4 independent experiments (±SEM). Representative histograms are shown. **b** Viability of cells described in (**a**) in response to phenformin (500 μM) and/or BSO (100 μM) treatment (upper graph) or phenformin (500 μM) and/or β-lapachone (4 μM) treatment (lower graph) for 48 h. The data are shown as fold change in viability compared to vehicle and is representative of n = 4 (upper graph) or n = 3 independent experiments (lower graph), presented as mean of means ± SEM. **c** Viability of BT474 and MDA-MB-231 cells treated with IACS-010759 (50 nM) and/or β-lapachone (BT474: 1.0 μM and MDAMB231: 0.5 μM) for 48 h. The data are shown as fold change in viability compared to DMSO, n = 4 independent experiments (mean of means) ± SEM (MDA-MB-231); or of n = 8 technical replicates, over two independent experiments ± SD (BT474). **d** DCFDA geometric MFI for MDA-MB-231 cells treated with IACS-010759 (50 nM) and/or β-lapachone (0.5 μM) for 24 h. The data are shown as fold change in MFI compared to DMSO and are representative of n = 3 independent experiments ± SEM. **e, f** Viability of HER2+ PDXs (CRC-132, GCRC2080) in vitro after treatment with phenformin (CRC-132, 100 μM; GCRC2080, 500 μM) alone and with (**e**) BSO (300 μM) or (**f**) β-lapachone (0.5 μM), for 48 h. The data are shown as fold change in viability compared to vehicle and is representative of n = 3 independent experiments (mean of means) ± SEM. **g, h** Viability of basal-like PDXs (GCRC1735, GCRC1915, GCRC1963, and GCRC1986) in vitro after treatment with **g** phenformin (500 μM) and/or BSO (300 μM) or **h** phenformin (500 μM) and/or β-lapachone (1 μM) for 48 h. The data are shown as fold change in viability compared to vehicle and are representative of n = 3 independent experiments (mean of means) ± SEM. P values were calculated using a two-way ANOVA with a Tukey's post hoc test. See also Supplementary Figures 9 and 10.

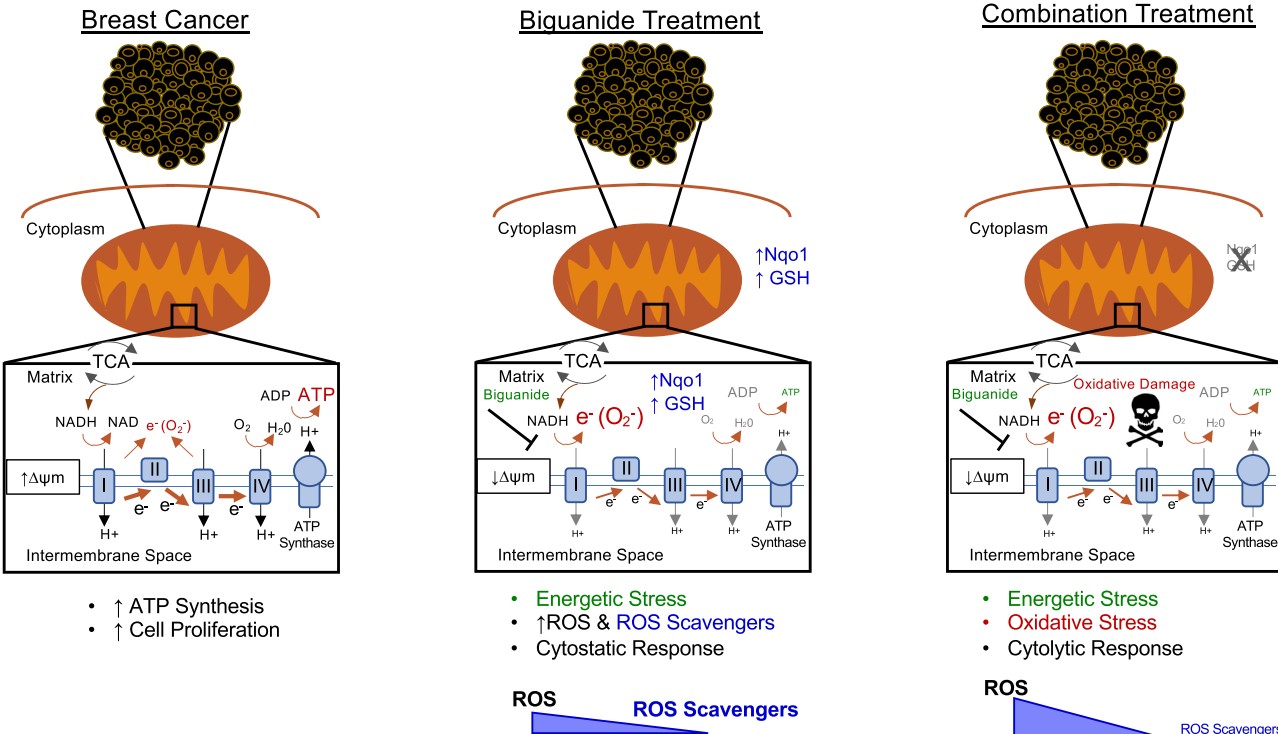

**Fig. 9 Targetable ROS-scavenging mechanisms selectively sensitize human breast cancers to biguanide treatment.** Most breast cancers are characterized by higher levels of oxidative phosphorylation and consequently increased mitochondrial membrane potential, in comparison to normal epithelial cells. Biguanides preferentially accumulate in cells with actively respiring mitochondria. Biguanide treatment as monotherapy inhibits complex I of the electron transport chain and OXPHOS leading to energetic stress. By inhibiting complex I, phenformin also increases mitochondrial superoxide generation. Combination therapy with phenformin and inhibiting tumor antioxidants, such as Nqo1 and glutathione, leads to oxidative stress in addition to energetic stress, and a potent tumoricidal response.

alleviate ROS defense mechanisms may also warrant further evaluation. Future research into the application of such combination approaches to treat multiple tumor types is supported by our results.

NQO1 is a classical *Nrf2* target gene with known ROS-scavenging properties. NQO1 catalyzes the two-electron reduction of quinones by utilizing NADH and NADPH as electron donors, preventing the development of ROS-generating unstable semi-quinones, that would otherwise be formed. NQO1 acts as a superoxide scavenger to maintain reduced forms of CoQ and vitamin E derivatives and modulates NAD(P)$^+$/NAD(P)H pools[66]. NQO1 further protects peroxisome proliferator-activated receptor gamma coactivator 1α (PGC-1α) from proteasomal degradation, a transcriptional coactivator and master regulator of genes that promote mitochondrial metabolism and ROS scavenging[67]. NQO1 is recognized as an attractive target in cancer as it is frequently overexpressed in tumors compared to normal tissues and increased NQO1 levels are strongly associated with late-stage disease and worse survival[43,44,68]. Indeed, NQO1-targeting drugs improve the efficacy of targeted therapies, including PARP inhibitors and immune checkpoint inhibitors, in a ROS-dependent manner[48,69]. Finally, several therapies, including ionizing radiation and chemotherapy, upregulate NQO1 levels in cancer cells[42]. PGC-1α levels are also increased by chemotherapies and elevated PGC-1α expression promotes resistance of breast cancer cells to mitochondrial complex I inhibitors[56]. Considering these observations, the extent to which STAT1-driven suppression of NQO1 expression relies on a concomitant loss of PGC-1α function warrants further investigation. Moreover, single-nucleotide polymorphisms (SNPs) in the *NQO1* gene, including the C609T mutation, result in decreased enzymatic

activity and have been associated with increased cancer susceptibility[70]. It is possible that tumors harboring such NQO1 SNPs would display increased sensitivity to phenformin, either as a monotherapy or in combination with drugs that target the glutathione system.

This study positions NQO1 as an important ROS scavenger that allows breast tumors to cope with phenformin, supporting previous research showing that NQO1 affords protection to the cytotoxic effects of rotenone, another mitochondrial complex I inhibitor[46]. We extend these findings to the novel small-molecule complex I inhibitor, IACS-010759, which also synergizes with β-lapachone. Thus, NQO1 is an attractive therapeutic target in oncology that is generalizable to multiple mitochondrial complex I inhibitors.

## Methods
**Cell culture.** Cells were grown in DMEM (MCF7, BT474, MDA-MB-231, BT20, MDA-MB-436, Hs578T, BT549, and 4T1-537) or RPMI (HCC1954) media supplemented with 10% fetal bovine serum (FBS), and 1× penicillin/streptomycin and 1× gentamicin (4T1-537 were supplemented with HEPES 10 mmol/L). MT4788, MT864, and their STAT1$^{-/-}$ counterparts (generated with CRISPR/Cas9)[29] were grown in Dulbecco's modified Eagle's medium (DMEM), 2.5% FBS, mammary epithelial growth supplement (MEGS: 5 mg/mL insulin, 3 ng/mL human epidermal growth factor, 0.5 mg/mL hydrocortisone, and 0.4% v/v bovine pituitary extract), and penicillin/streptomycin and gentamicin. The NIC cell line was generated as described[50] and cultured in DMEM supplemented with 5% FBS and MEGS, and penicillin/streptomycin and gentamicin. NMuMG and NT2197 cells were described[50] and grown in DMEM supplemented with 10% FBS, 10 μg/mL insulin, 10 mmol/L HEPES, and penicillin/streptomycin and gentamicin. NOP6, NOP23 (graciously provided by Dr. Brad Nelson)[71] were cultured in DMEM, 5% FBS, 1× insulin, transferrin and sodium selenite (ITSS) (Sigma), and with penicillin/streptomycin and gentamicin. PDX cell lines GCRC2080, GCRC1735, GCRC1971, GCRC1986, and GCRC1963 were grown in F media: 3:1 DMEM (Wisent):F12 Nutrient Mixture (Wisent), 5% fetal bovine serum, hydrocortisone 25 ng/mL, insulin 5 μg/mL, cholera toxin 8.4 ng/mL (Sigma), epidermal growth factor

(Invitrogen) 0.125 ng/mL, gentamicin 50 μg/mL, and Y-27632 (Enzo Life Sciences) 10 μmol/L[53]. PDX CRC-132 cell lines was grown in 66% DMEM high glucose, 25% F12 nutrient mixture (Gibco), 7.5% FBS, 10 μM Rock inhibitor, 10 ng/mL epidermal growth factor, 8.4 ng/mL cholera toxin, 5 μg/mL insulin, 0.4 μg/mL hydrocortisone, 1.48 mM L-glutamine, and penicillin/streptomycin[52]. All cell lines were grown in 37 °C, 5% $CO_2$, and minimally screened for mycoplasma infection monthly or 24 h prior to any in vivo injection using the MycoAlert™ mycoplasma detection kit (Lonza).

**PDX cell lines**. PDXs were previously developed from tumor material graciously donated by patients who provided informed consent[52,53]. The procedures were in accordance with the McGill University Health Center research (SUR-99-780); Jewish General Hospital ethics boards for (1) JGH breast biobank (protocol # 05-006) and (2) the generation of patient-derived material protocol (14-168). For the Goodman Cancer Research Centre (GCRC) PDX cell lines (GCRC2080, GCRC1735, GCRC1915, GCRC1963, and GCRC1986) tumor fragments taken from mice were minced and digested in a rotator shaker at 37 °C for 1 h with an enzymatic mix of one part collagenase-IV, nine parts of Digestion Media (DMEM, FBS, HEPES, and gentamicin). Tissue was later digested with trypsin 0.25% and a mix of DNaseI (10 μL)/dispase (1 mL). Murine cells were removed using a Mouse Cell Depletion Kit (Miltenyi), and single-human epithelial cancer cells were culture in their respective F media as described above. For CRC-132, tumor fragments from PDX-132 were incubated with a mix of collagenase/hyaluronidase and dispase (STEMCELL Technologies) for 1 h at 37 °C on an oscillator to perform tissue dissociation. After centrifugation at $200 \times g$ for 5 min and resuspension in DMEM 10% FBS, cells were filtered through a 70 μM cell strainer, centrifuged at $200 \times g$ for 5 min, and resuspended in F medium. Cells were then transferred to a T25 flask containing lethally irradiated 3T3-J2 cells ($1 \times 10^6$ cells). After five passages, coculture with irradiated 3T3-J2 cells was suspended, murine cells were removed using a Mouse Cell Depletion Kit (Miltenyi), and cells were grown in the conditioned medium (three parts of conditioned medium for one part of fresh medium) for another ten passages and then maintained in F-medium.

**Animal models**. FVB, SCID-Beige, and BALB/c female mice (7–10 weeks old) used for mammary fat pad injections were purchased from Charles River Laboratories (Quebec, Canada). Female IFNγ$^{-/-}$ and CD8$^{-/-}$ (7–10 weeks old) were previously backcrossed onto an FVB background[29] and also used for mammary fat pad injections. Mice were age matched within an experiment between groups. Female NSG mice were previously used for PDXs (The Jackson Laboratories, Strain # 005557). All mice had ad libitum access to food and water and housed within the animal facilities of the Lady Davis Institute, on a 12 h light day cycle, mean temperature $22.5 \pm 1.5$ °C, and 22–28% humidity. These studies were approved by and follow the Animal Resource Centre at McGill University procedures (protocols: 2011-5864, 2014-7514, and 2001-4830). These experiments comply with the guidelines set by the Canadian Council of Animal Care.

**Mammary fat pad injections**. Cells [$0.05 \times 10^6$ (4T1-537; Balb/c); $0.5 \times 10^6$ (MT4788; FVB); or $1 \times 10^6$ cells (MDA-MB-231; SCID-Beige)] were injected into the fourth mammary fat pads of anesthetized mice. For MDA-MB-231 and 4T1-537 injections, as well as for the oncolytic virus study, cells were resuspended in a 1:1 PBS and Matrigel (Corning) mixture. Otherwise, cells were resuspended in sterile PBS. Tumors volumes were measured by digital caliper every 2 days using the equation: volume $= 4/3 \times (3.14159) \times (length/2) \times (width/2)^2$. Drug treatment studies were initiated when tumors reached an initial volume of ~100–150 mm$^3$.

**Drug preparation and treatment**. Phenformin hydrochloride and metformin hydrochloride (Cayman Chemicals) powder was dissolved in PBS, filter sterilized, and stored for a maximum of 4 weeks at 4 °C. For in vitro experiments, phenformin was used at a final concentration ranging from 20 to 500 μM. For the in vivo studies, phenformin was administered intraperitoneally at a concentration of 50 mg/kg daily (unless specified).

β-Lapachone (Cayman Chemicals) was dissolved in dimethyl sulfoxide (DMSO) at a concentration of 100 μM (24.27 mg/mL) and frozen at −80 °C for up to 4 months. For the in vitro experiments, β-lapachone was used at a final concentration of 0.5–4 μM. DMSO served as the vehicle control. For in vivo experiments, β-lapachone was dissolved in 225 mg/mL hydroxypropyl-β-cyclodextrin (HPβCD) (Cayman Chemicals) in sterile PBS, protected from light, and heated to 70 °C for $3 \times 10$ min. β-Lapachone/HPβCD was stored at room temperature for up to 2 weeks[47]. Mice were treated with 25 mg/kg β-lapachone-HPβCD or with 225 mg/mL HPβCD/PBS (vehicle control), every 2 days, intraperitoneally. For in vivo studies, β-lapachone-HPβCD and HPβCD/PBS treatment were started 2 days prior to phenformin (or PBS) treatment. BSO (Cayman Chemicals) was dissolved in PBS heated to 37 °C. MitoTEMPO (Sigma) stock was prepared in DMSO (allowing for longer storage) or in PBS (2 weeks at 4 °C). For the in vitro studies, MitoTEMPO was used at a final concentration ranging from 5 to 10 μM. Cells were pretreated with MitoTEMPO for 24 h prior to the start of drug treatment and MitoTEMPO-containing media were changed every 24 h thereafter. For the in vivo experiment, mice were treated with 3 mg/kg MitoTEMPO (or PBS control).

Mouse and human recombinant IFNγ (R&D Systems) were resuspended in PBS and stored at −80 °C. The final concentration of IFNγ was 1 ng/mL unless specified. PolyIC high molecular weight form (InvivoGen) was prepared as per the manufacturer's protocol and stored at −20 °C. For the in vivo experiments, mice were treated with 50 μg of polyIC per mouse, every 2 days, intraperitoneally (50 μL), or with saline control. PolyIC treatment was started 2 days prior to the start of phenformin treatment. This dose of polyIC was selected based on a previous report[72]. For the immune checkpoint inhibitor studies, 100 μg of neutralizing anti-PD1 antibody (clone RMP1-14, BioXCell) was injected intraperitoneally every 3 days. Isotype control IgG (InVivoMAb rat IgG2a, clone 2A3, BioXCell) was injected using a similar dosing schedule. For the oncolytic virus VSV (MΔ51) studies, the virus was administered through two consecutive intratumoral injections of $1 \times 10^7$ particle forming units (PFUs) per tumor in total volume of 50 μL PBS, administered 24 h apart. PBS was injected intratumorally as the control.

**In vitro viability cell counts**. Cells were incubated with various drugs and/or with media containing different concentrations of glutamine and glucose (as indicated in the figure legends) for the times indicated. For the glucose or glutamine deprivation studies, cells were cultured in glucose-free DMEM media (319-061-CL) supplemented with 1 mM sodium pyruvate or glutamine-free DMEM (319-025-CL) that were combined with complete DMEM media (319-005-CL) to attain final glucose or glutamine concentrations. Cells were trypsinized and resuspended in media and live cells were quantified by trypan blue exclusion using a hemocytometer.

**Immunohistochemistry**. Tumor pieces were fixed in 10% neutral buffered formalin immediately after euthanasia for 18–24 h at room temperature and stored in 70% ethanol at 4 °C until paraffin embedding. Paraffin-embedded sections (5 μm) were then subjected to IHC staining. Antigen retrieval was performed in 10 mM sodium citrate buffer in distilled water, pH adjusted to 6.0 with 1 N HCl, supplemented with 0.05% Tween-20 in a pressure cooker for 12 min, and cooled on ice for 30 min. Slides were then washed $2 \times 5$ min with TBST (Tris-buffered saline/0.05% Tween-20), rinsed 2× in TBS, and blocked for 10 min with unconjugated avidin followed by unconjugated biotin (BioLegend). After another 5 min TBST wash, slides were blocked with 10% bovine serum albumin (BSA)/TBS, and then incubated with the primary antibody in 2% BSA/TBS overnight at 4 °C. Specific information regarding the primary antibodies used can be found in Supplementary Table 1. Slides were subsequently washed in TBST ($3 \times 5$ min), blocked with freshly diluted 3% hydrogen peroxide, washed $2 \times 5$ min, and incubated with the appropriate biotinylated secondary antibody in 2% BSA/TBS for 1 h at room temperature. Slides were again washed in TBST ($3 \times 5$ min), rinsed 1× TBS, and incubated for 30 min with avidin/biotinylated complex reagent (Vectastain®, Vector Laboratories). This was followed by timed incubation with DAB reagent (Vector Laboratories) for development, which was stopped with tap water. For the mouse antibody 8-oxo-dG antibody, there was no antigen retrieval step and the slides were instead stained using Mouse on Mouse Polymer kit (Abcam), as per the manufacturer's protocol. All slides were then dehydrated, counterstained with 20% hematoxylin (Fisher Scientific), and mounted with ClearMount™ media (StatLab). For the 8-oxo-dG antibody, there was no antigen retrieval step and the slides were stained using a Mouse on Mouse Polymer kit (Abcam), as per the manufacturer's protocol. All slides were then dehydrated, counterstained with 20% hematoxylin (Fisher Scientific), and mounted with ClearMount™ media (StatLab). Slides were scanned with ScanScope XT Digital Slide Scanner (Aperio). Images were analyzed with the ImageScope software (Aperio) using positive pixel count or nuclear algorithms.

**Immunoblot analysis**

*Whole-cell lysates*. Whole-cell lysates were prepared by lysing cells with PLCγ buffer (50 mM HEPES [pH 7.5], 150 mM NaCl, 10% glycerol, 1% Triton X-100, 1 mM EGTA [pH 8.0], 1.5 mM MgCl$_2$, 5 mM NaVO$_4$, 5 mM NaF, and PIN: 1 μg/mL chymostatin, 2 μg/mL antipain, 2 μg/mL leupeptin, 1 μg/mL pepstatin, 2 μg/mL aprotinin) for 10 min on ice.

*Tumor lysates of lung cancer brain metastases*. The PDX models of lung cancer brain metastases used in this study were previously established[45,73] and were approved by McGill University and the Montreal Neurological Institute Hospital ethics boards (MNIH) (IRB # 2018-4150). Surgically resected brain metastasis patient material was received from the neurosurgery operating room at the MNIH. Tumor fragments were expanded as patient-derived xenografts in the subcutaneous flank of NSG mice. Once tumors reached a size of >250 mm$^3$, mice were euthanized and tumors were flash frozen in liquid nitrogen. Flash-frozen tumor pieces were crushed in liquid nitrogen and then lysed in 200–500 μL RIPA buffer (10 mM Na phosphate [pH 7.0], NaCl 150 mM, NP-40 1.0%, sodium dodecyl sulfate (SDS) 0.1%, Na deoxycholate 1.0%, NaF 10 mM, EDTA 2 mM, 5 mM NaVO$_4$, PIN: 1 μg/mL chymostatin, 2 μg/mL antipain, 2 μg/mL leupeptin, 1 μg/mL pepstatin, 2 μg/mL aprotinin), and mixed by pipetting multiple times.

Both whole-cell and tumor lysates were centrifuged at $16,000 \times g$, 4 °C for 10 min. Protein concentration was measured by Bio-Rad Protein assay. Lysates were then separated by SDS-polyacrylamide gel electrophoresis (SDS-PAGE) and

transferred onto polyvinylidene difluoride membranes. Membranes were then blocked in 3% bovine serum albumin (BioShop) or 5% milk in TBST and probed with antibodies as listed in Supplementary Table 2. Secondary antibodies conjugated to horseradish peroxidase (1:10,000) and ECL (Thermo Fisher) or Luminata Forte HRP substrate (Millipore Sigma) were used for protein detection. Uncropped versions of scanned films can be found in the Source data.

**Flow cytometry.** Cells were treated with various drug combinations as outlined (see figure legends). After the specific staining protocol outlined below, cells were analyzed with the BD LSR Fortessa. Appropriate single stained and unstained controls were used for each experiment. Analysis was performed with the FlowJo Software, version 10.

*General oxidative stress indicator.* A total of $1 \times 10^6$ cells in suspension were incubated with CM-H2DCFDA (Invitrogen) (0.5–5 μM, concentration determined through titration experiments) in PBS for 30 min at 37 °C and protected from light. Cells were washed and stained with a final concentration of 0.25 μg/mL PI (BD Biosciences) for 15 min, protected from light, prior to flow cytometric analysis with the BD LSR Fortessa.

*MitoSOX^{TM} Red mitochondrial superoxide indicator.* A total of $5 \times 10^5$ cells in suspension were incubated with MitoSOX^{TM} (Invitrogen^{TM}) for 10 min at 37 °C and protected from light. Cells were then washed, stained with a final concentration of 2.5 μg/mL 4′,6-diamidino-2-phenylindole (Vector laboratories) for 10 min, and protected from light, prior to flow cytometric analysis with the BD LSR Fortessa.

*Proliferation assay.* Cells were cultured with BrdU (0.5 μL/mL of media) for 18 h prior to the end of the experiment. Cells were trypsinized, washed with 3% BSA/ PBS, and stained as per the Phase-flow^{TM} BrdU-Alexa Fluor® 647 proliferation kit for Flow Cytometry (BioLegend), as per the manufacturer's protocol. Specifically, cell pellets were fixed by resuspending in 100 μL Buffer A at 4 °C for 20 min, and then washed with 3% BSA/PBS. Cell pellets were resuspended in 1 mL of 90% FBS–10% DMSO, and then stored at −80 °C overnight. Cells were then thawed at 37 °C, counted, and $1 \times 10^6$ cells were aliquoted and washed with 2 mL of 1× Buffer B and centrifuged for 5 min at $200 \times g$. Wash was carefully aspirated, leaving ~50 μL of liquid in each tube. Cell pellets were then permeabilized by resuspending in 100 μL of Buffer C and incubating at room temperature for 10 min. A repeated wash step with 1 mL 1× Buffer B was performed. Cell pellets were then fixed a second time by gently resuspending cells in 100 μL of Buffer A and incubating for 5 min at room temperature. The wash step was repeated as above and the supernatant was discarded. Cells were next incubated with 50 μL of DNase (400 μg/mL stock) at 37 °C for 1 h and then stained with the anti-BrdU antibody-Alexa Fluor®-647 for 15 min at room temperature in the dark, and then washed as above. Cells were then resuspended in 100 μL of PBS and samples were analyzed with the BD LSR Fortessa flow cytometer.

*Apoptosis assay.* Cells were first washed in PBS and counted. After centrifugation, cell pellets were resuspended in Annexin V binding buffer (BD Pharmingen) at $1 \times 10^5$ cells/100 μL. One hundred microliters of this cell suspension was stained with 5 μL of Annexin V Alexa Fluor® 647 (BioLegend) antibody for 15 min and then add 0.25 μg/mL PI (BD Biosciences) to the tube and incubated in the dark for 15 min. Samples were analyzed by flow cytometry with BD LSR Fortessa.

**NQO1 knockdown and overexpression.** HEK293T cells were transfected with 2 μg shRNA constructs and 2 μg packaging plasmids PsPAX2, PMD2.G (Addgene) using calcium phosphate precipitation. Media were changed and virus-containing media were collected after 12 h. After replenishing the media, 12 h later, the lentiviral supernatant was collected again 12 h later. MT4788, BT474, and MDA-MB-231 cells were retrovirally infected with shRNAs targeting mouse or human NQO1 or non-mammalian shRNA control plasmid DNAs (see Supplementary Table 3 for nucleotide sequences). shRNA-containing plasmids were obtained through McGill's Genetic Perturbation Service and then selected with puromycin (Thermo Fisher Scientific). For the overexpression studies, the mouse NQO1 complementary DNA (cDNA) was PCR amplified from NQO1 cDNA ORF clone expression plasmid (Sino Biological) and inserted into pQCXIP plasmid (Clontech) via *Not*I and *Eco*RI sites added during amplification (forward: AAAGCGGCCGCATGGCGGCGA GAA; reverse: CCCCCCCGAATTCTTATTTTCTAGCTTTGATCTGGTTG).

**RT-qPCR analysis.** Total RNA was isolated from cell lines grown in 6-well plates, using TRIzol reagent (Invitrogen) according to the manufacturer's protocol. RNA concentration was determined using SynergyMIX. With input RNA concentration consistent among all samples, complementary DNA was synthesized with SSII (Life Technologies) as per the manufacturer's protocol, using Random Primers (New England Biolabs) or with 5× All-in One RT MasterMix (Applied Biological Materials Inc.). Quantitative RT-PCR was performed with EvaGreen 2× qPCR mixture (Diamed) and primer sequences listed in Supplementary Table 3.

**RNA-sequencing.** MT864 and MT4788 (STAT1-WT and STAT1-KO) cell lines were cultured for 24 h with 1 ng/mL IFNγ and total RNA was extracted using RNeasy Midi Kits (Qiagen). RNA-seq was performed at the McGill University and Genome Quebec Innovation Centre. RNA quality was assessed by Agilent 2100 Bioanalyzer (Agilent Technologies). Libraries for RNA-seq were prepared according to strand-specific Illumina TruSeq protocols. Samples were multiplexed at four samples per lane and sequenced on an Illumina HiSeq 2500 PE125 instrument.

Raw reads were trimmed using Trimmomatic v0.32[74]. First, adaptors and other Illumina-specific sequences from each read were removed using palindrome mode. Then, a four-nucleotide sliding window removes the bases once the average quality within the window falls <30. Next, the first four bases at the start of each read were removed. Finally, reads shorter than 30 bp were dropped. Cleaned reads were aligned to the mouse reference genome build mm10 using STAR v2.3.0e[75] with default settings. Reads mapping to >10 locations in the genome (mapping quality < 1) were discarded. Gene expression levels were estimated by quantifying uniquely mapped reads to exonic regions (the maximal genomic locus of each gene and its known isoforms) using feature Counts[76] (v1.4.4) and the Ensembl gene annotation set. Normalization (mean of ratios) and variance-stabilized transformation of the data were performed using DESeq2[77] (v1.14.1). Multiple control metrics were obtained using FASTQC (v0.11.2), samtools[78] (v0.1.20), BEDtools[79] (v2.17.0), and custom scripts.

**Gas chromatography/mass spectrometry (GC/MS) and metabolite extraction.** Cells at ~80% confluency were washed twice in cold saline solution (NaCl, 0.9 g/ l) and then quenched with 600 μL of 80% iced methanol on dry ice. Following 10 min of sonication on slurry ice using a bath sonicator (Bioruptor) with the cycling 30 s on/ off at the highest settings, the homogenates were centrifuged at $14,000 \times g$ at 4 °C for 10 min. Supernatants were collected and supplemented with 750 ng of myristic acid-D_{27} (an internal standard; Sigma-Aldrich, ON, Canada) and dried overnight in a cold vacuum centrifuge (Labconco). The dried samples were reconstituted with 30 μL of methoxyamine-HCl (10 mg/mL dissolved in pyridine; Sigma) and incubated for 30 min at room temperature. Next, the samples were derivatized with MTBSTFA (Sigma). After incubation for 1 h at 70 °C, 1 μL of each derivatized sample was injected into the GC/MS instrument (5975C, Agilent). Data were acquired in scan mode and analyzed with the MassHunter software (Agilent) as described[28,80]. The level of each metabolite was normalized by the intensity of myristic acid-D_{27} and the average cell number of three independent wells per treatment (run in parallel), on the day of each separate biological repeat experiment.

**Seahorse respiration assay.** Cells were seeded, cultured, and treated in Seahorse XF24 cell culture microplates (Agilent, ON, Canada). The ECAR and the OCR were determined using the Seahorse XFe24 Analyzer and Wave Desktop Software (Agilent). The cartridge was incubated at 37 °C with a calibrant, overnight. On the day of the assay, cells were washed twice and then incubated with supplemented Seahorse XF base medium (Agilent) [glucose (25 mM), L-glutamine (4 mM), and sodium pyruvate (1 mM), adjusted to pH 7.4 and filter sterilized 0.2 μM], in a CO_2-free incubator at 37 °C. After 1 h and calibration of the sensor cartridge, the plate was loaded in the XFe24 Analyzer and the bioenergetics was determined following injection of 1 μM oligomycin, 1 μM FCCP (fluoro-carbonyl cyanide phenylhydrazone), 0.5 μM rotenone/antimycin A, and 20 μM monensin. The rates of ATP production (by glycolysis, $J_{ATP}$ glycolysis, and by OXPHOS, $J_{ATP}$ oxidative) were calculated by adding monensin, as described[81]. Measurements were normalized to the number of cells, as determined by parallel cell counts for each treatment on the day of each biological repeat.

**Liquid chromatography/mass spectrometry.** Cells were washed three times with 150 mM ice-cold ammonium formate and scrapped on dry ice using two different conditions: (1) for glutathione, cells were scraped followed by the addition of 50% high-performance liquid chromatography (HPLC)-grade methanol, 220 μL of ice-cold acetonitrile (Fisher) were added to the slurry and then homogenized using the beat beater. After bead beating, 600 μL of ice-cold dichloromethane (Fisher) and 300 μL of HPLC-grade water was then added to the homogenates, following 10 min centrifugation at $1500 \times g$ at 1 °C. The upper aqueous phase was transferred to a new tube and left to dry overnight by vacuum centrifugation with sample temperature controlled at −4 °C (Labconco). (2) NADH/NAD^{+}/NADPH/NADP^{+} extraction cells were scraped into 600 μL 80% HPLC-grade methanol and allowed to rest at −80 °C overnight. Samples were then homogenized by bead beating. A volume of 600 μL of ice-cold dichloromethane (Fisher) and 300 μL of HPLC-grade water were then added to the homogenates, followed by 10 min centrifugation at $1500 \times g$ at 1 °C. The upper aqueous phase was transferred to a new tube and left to dry overnight as above. All targeted analyses were carried out on an Agilent 6430 Triple quadrupole QQQ; 1290 Infinity ultra-performance LC System equipped with a Scherzo SM-C18 column 3 μm, $3.0 \times 150$ mm$^2$ (Imtakt Corp, Japan) at 10 °C. Multiple reaction monitoring (MRM) transitions were optimized on authentic standards. Data were quantified by integrating the area under the curve of each compound using MassHunter Quant (Agilent). Relative concentrations were determined from external calibration curves. No additional corrections were made for ion suppression or enhancement, thus relative metabolite responses are presented.

For GSH and GSSG measurements, dried samples were solubilized in 35 μL HPLC-grade $H_2O$ and 25× dilutions prepared. A volume of 5 μL injected into LC-MS where GSH and GSSG were chromatographically separated at a flow rate of 0.4 mL/min by starting with 100% solvent A (0.2% formic acid in water) for 2 min. The gradient was increased to 80% solvent B (0.2% formic acid in methanol) over a period of 6 min. Solvent B was increased to 100% for column washing for 5 min and then re-equilibrated to 100% A for 6 min before the next injection. The electrospray ionization (ESI) source and samples were analyzed in positive ionization mode. MRM transitions quantifier and qualifier ions were, respectively, $308.1 \rightarrow 179.0$ and $308.1 \rightarrow 76.0$ for reduced glutathione, and $613.2 \rightarrow 355.1$ and $613.2 \rightarrow 231.0$ for oxidized glutathione. Ion source gas temperature and flow were set at 350 °C and 10 L/min, respectively, nebulizer pressure was set at 40 psi and capillary voltage was set at 3500 V.

For NAD, NADH, NADP, and NADPH measurements, chromatographic separation was achieved using a chromatographic gradient started at 100% mobile phase A (50 mM ammonium acetate/50 mM $NH_4OH$: 9/1, pH 8.6) for 2 min followed by an 8 min gradient to 40% B (100 mM ammonium acetate/100 mM $NH_4OH$: 9/1, pH 8.6)/ACN: 80/20) at a flow rate of 0.4 mL/min. This was followed by a 5 min hold time at 100% mobile phase B and a subsequent re-equilibration time (6 min) before thhe next injection. Individual samples were resuspended and run immediately to minimize the loss of NADH and NADPH. A sample volume of 10 μL was injected. The mass spectrometer was equipped with an ESI source and samples were analyzed in positive ionization mode. MRM transitions quantifier and qualifier ions were, respectively, $664.1 \rightarrow 135.9$ and $664.1 \rightarrow 428.1$ for NAD and $666.1 \rightarrow 514.0$ and $666.1 \rightarrow 136.0$ for NADH. Ion source gas temperature and flow were set at 350 °C and 10 L/min, respectively, nebulizer pressure was set at 40 psi and capillary voltage was set at 3500 V.

**Statistical analysis**. Statistical analyses and generation of graphs were performed using the GraphPad Prism (v6 and v9) software, or Microsoft Excel (v16.46) (Fig. 3j and Fig. S3e), see details on statistical tests performed in figure legends. Flow cytometry analyses and statistics (geometric fluorescent mean and percentages) were determined with the FlowJo Software 10.

**Reporting summary**. Further information on research design is available in the Nature Research Reporting Summary linked to this article.

## Data availability
RNA-sequencing data that support the findings of this study have been deposited in Gene Expression Omnibus with the accession code GSE153189. Mouse reference genome mm10 used in this study can be accessed using http://ccb.jhu.edu/software/tophat/igenomes.shtml (under Mus Musculus/UCSC/mm10). Uncropped versions of scans of western blots can be found in the "Source data" section. Supplementary Data files 1, 2, and 3 have been provided for the RNA-seq analyses presented in Fig. 6a and Supplementary Figure 6a–c. Further information and requests for resources and reagents should be directed to and will be fulfilled by the corresponding author. Source data are provided with this paper.

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

## Acknowledgements

We gratefully acknowledge Dr. Brad Nelson who generously provided the NOP cell lines used in this study. We also thank our colleagues in the George and Olga Minarik Research Pathology Facility (Lady Davis Institute) and the McGill Histology Core Facility (Goodman Cancer Research Centre) for the preparation of tissue for histological analysis as well as the Small Animal Research and Flow Cytometry Cores (Lady Davis Institute) for their support with the animal studies and the flow cytometry experiments, respectively. Finally, we thank the Goodman Cancer Research Centre (GCRC) Metabolomics Core Facility for all the metabolomics studies. This project was funded by CIHR operating grants to J.U.-S. (#111143) and J.U.-S. and J.S.-P. (#244105) and funding from Réseau de Recherche sur le Cancer of the FRQS, Québec Breast Cancer Foundation, and Oncopole (to Morag Park). The JGH Breast biobank is supported by FRQS Réseau Recherche Cancer and Quebec Breast Cancer Foundation. P.M.S. acknowledges support from a Terry Fox program project grant. S.P.T. holds a CIHR doctoral award, Y.K.I. and R.L.S. were funded by an FRQS doctoral studentship, and M.D. holds a Vanier Canada Graduate award (CIHR). S.P.T. and M.D. also wish to acknowledge support from McGill's M.D.-Ph.D. program stipend. N.S.M. would like to acknowledge the Fundación para la Salud y la Educación Dr. Salvador Zubirán and FOINS-INCMNSZ postdoctoral fellowship (A-307-7). J.U.-S., M.W., C.L.K., and I.T. are FRQS Research Scholars, J.S.-P. holds a CRC Tier I award, P.M.S. is a William Dawson Scholar, and MPark is a James McGill Professor and holds the Diane and Sal Guerra Chair in Cancer Genetics at McGill University.

## Author contributions

S.P.T., M.W., R.L., C.L.K., J.S.-P., and J.U.-S. conceived and designed the experiments. S.P.T., Y.K.I., E.C.C., O.N., A.N., S.H., R.A., K.L., B.L., and J.U.-S. collected and/or analyzed the data. R.L.S., V.S., C.M., P.S., H.K., D.A., N.S.M., C.C., A.A.-M., M.-L.G., M.D., K.P., M.B., Michael Pollak, I.T., P.M.S., C.L.K., Morag Park, and J.S.-P. provided essential reagents and expertise. S.P.T. and J.U.-S. wrote the manuscript.

## Competing interests

The authors declare no competing interests.
