## [Peer Review File · Nature Communications]

REVIEWER COMMENTS

Reviewer #1 (Remarks to the Author):

This manuscript probes the therapeutic utility of strategies to simultaneously induce ROS (as in mitochondrial complex I inhibition via phenformin) and inhibit cellular ROS scavenging capacity in breast cancer. The authors present multiple in vivo studies demonstrating that several clinically relevant inhibitors of cellular antioxidant mechanisms sensitize breast cancer models to the anti-tumor effects of phenformin. They also identify transcriptional inhibition of the ROS scavenging enzyme NQO-1 as a previously unappreciated mechanism whereby tumor inflammation via IFN γ -STAT1 signaling amplifies oxidative stress, sensitizing tumors to phenformin. Overall, the findings reported here have important translational implications, however the biochemical/metabolic mechanism underlying the sensitizing effect of the IFN γ /STAT1/NQO-1 pathway has not been sufficiently worked out.

Comments:

1. Manipulation of ShcA is predicted to affect several signaling/transcription events beyond STAT1 (including STAT3 inhibition as the authors have previously shown, PMID:28276425), which can explain changes in the sensitivity of ShcA Y313F cells to phenformin. The authors should provide a stronger rationale for the focus on STAT1 in this system. This is relevant given the importance of STAT3 in metastatic capacity, immune tolerance and rewiring fuel metabolism in breast cancer (PMID: 30100196, 29249690), as well as its capacity to activate antioxidant gene expression (PMID: 21715323). Inhibition of STAT3 will be consistent with many of the cellular phenotype the authors observe.
2. The spectrum of inflammatory signaling induced by PolyIC is much broader than STAT1/IFN γ . It would be important to provide evidence that the observed anti-tumor effect Fig. 2C is specifically mediated by STAT1. A strong direct test for the role of STAT1 in the proposed mechanism of in vivo sensitization to phenformin is to test tumor formation in STAT1 KO mice injected as in Fig. 2A or 2C.
3. Metabolic profiling of IFN γ -treated cells in figure 3 requires additional depth. The authors conclude IFN γ has minimal effect on bioenergetic parameters, however, changes in fuel metabolism and redox balance can still be relevant. Changes in steady state levels of certain metabolites suggest multiple possibilities some of which are described in the text but these should be directly tested. This is relevant given the extent of published literature on metabolic characteristics of breast cancer and will help advance the field on metabolic mechanisms underlying the sensitizing effect of IFN γ /inflammation on phenformin tumoricidal capacity.
4. Related to the above point, it will be important to perform the metabolic characterization in the context of IFN γ and phenformin in addition to IFN γ -only treatment. This is relevant for a deeper understanding of the metabolic understanding of IFN γ 's sensitizing effects.
5. Overall, the main figures will benefit from some reorganization to more effectively package key data and eliminate figure panels that are not absolutely relevant to the main discovery of the paper. Examples include (but are not limited to) Fig. 1G and H, which can be moved to supplements.
6. The insight gained from ECAR data in Fig. 3C is unclear. How do the authors interpret the

ECAR data vis-à-vis changes in lactate and pyruvate. See also point 3 above.

Reviewer #2 (Remarks to the Author):

In this manuscript, the authors address a number of issues, mainly surrounding the potential of modulating reactive oxygen species as a therapeutic strategy for breast cancers. Focusing largely on phenformin and the role of STAT1-repressed NQO1, they show that in a wide variety of model systems phenformin can potentiate a variety of disparate therapies, particularly showing strong combinatorial effects with other treatments affecting ROS metabolism.

This paper is characterized by an abundance of experimental data in a wide variety of systems. There are, in fact, so many different systems used with unclear rationales, that it is difficult to crystallize specific high-impact findings. For example, data are presented on combinations of phenformin with an oncolytic virus or with an anti-PD-1 antibody, which have almost nothing to do with the rest of the data, and the material on b-lapachone and IACS-010759 is similarly very tangential. A tighter focus on the effects of STAT1 on NQO1, and how this synergizes with phenformin (or other related compounds), as indicated by the title of the manuscript, would likely be much more effective. In addition, the following points should be considered:

1. In the sections on the role of interferon-g and STAT1 in mediating effects with phenformin (and NQO1), the concepts of increased STAT1 expression, STAT1 phosphorylation, and STAT1 transcriptional activity are used intermittently, without critically assessing their relationships to each other. Since transcriptionally active STAT1 can increase STAT1 transcription, increased STAT1 levels can reflect increased transcriptional activity. However, it is not clear if the authors are trying to make this point. Similarly, STAT1 is thought to be transcriptionally active only when phosphorylated on tyrosine-701, though counter-examples have been proposed. Since these are critical points to understand the connection with phenformin sensitivity and NQO1 repression, they should be addressed directly.
2. The second section of the results concerns the effect of “inflammation” on sensitizing breast cancer models to the tumoricidal effects of phenformin. While this seems to relate to findings with Poly IC, it is not clear how the authors are defining inflammation, as there are many contradictory findings here. In fact, the authors state that the “observed cooperativity is likely distinct from metabolic stress or inflammatory processes.” A subsequent section is entitled, “Inflammation-induced phenformin sensitivity requires mitochondrial ROS.” Here, inflammation seems to be referring to IFN-g treatment (which is likely not “inflammatory” in these systems in any conventional sense). Even the term “STAT1-induced inflammation” in the title seems imprecise given the data in the paper (and it is not clear that STAT1 can “induce” inflammation). This section in particular, and the manuscript in general, would benefit from more precision in the use of scientific terms.
3. The issue of the known human pharmacokinetics of these drugs should be addressed. One of the reasons that metformin (and potentially phenformin) have not performed as well in human studies as in cell culture and murine studies is that the therapeutic levels achieved in people are much lower. This issue should at least be noted in this manuscript.

Reviewer #3 (Remarks to the Author):

In this paper, the authors explored a potential combination strategy using both IFN γ -STAT1 activator and ROS generator phenformin for breast cancer therapy. They demonstrated that activated STAT1 reduces the NQO1 level and potentiates oxidative stress to sensitize tumor cells to mitochondrial complex I inhibitor. This study reveals a novel vulnerability of breast tumors to combinations of phenformin with inhibitors that hinder anti-oxidant defense mechanisms. The findings are very interesting and will provide the important knowledge to design new drug strategy in oncology. Although the authors have presented the extensive data, a number of concerns given below still need to be addressed to improve the whole manuscript.

1. In the Western blot data for some cell lines, it looks like IFN γ -STAT1 has regulation effects on Tubulin expression. The authors may need to be careful about this internal control pick. For example, the expression of Tubulin reduced in 6738 cells that were stimulated by IFN- γ (see Figure 1B) and in MT4788 cells with STAT1 KO (see Figure 1G).
2. In Figure 1, when the authors tested the cooperative effects of IFN γ and phenformin in different breast cancer cell lines, they did not test any ER+ cell lines such as MCF7. Can the authors give me an explanation for not doing tests in ER+ breast cancer lines?
3. The data in Figure 1I panel are showing some conflict: IFN- γ treatment alone had no effect on cell viability of MT4788 (STAT1-WT). But IFN- γ treatment alone increases cell viability of MT4788 (STAT1-KO)? An explanation is needed for this phenomenon.
4. For some reason, when the authors collected the tumor volume data under different treatments for the different panels in Figure S1 and Figure 2, they used different termination days (8, 10 and 13) for different panel. I feel the results were completely influenced by the termination time pick. For example, why the authors chose DAY10 not DAY8 in Figure S1A but DAY8 in Figure 2A-C? I believe the different timing pick will cause bias to make the right conclusion. For example, I can conclude that there is no obviously additive effect on DAY8 in Figure S1A. Moreover, Figure S1B missed the data on DAY8 for some reason (it could be explained as the PFU of oncolytic virus was too high so that the tumor progression was almost completely suppressed at the first 6 days of treatment and no further effect in the combination group at last). There should be an additive effect in the combination group because the mechanisms of oncolytic virus and Phenformin must be different.
5. The connection of the anti-tumor mechanism of phenformin to apoptosis needs to be more careful for some data. There is no difference of Ki67 expression or Casp3 cleavage after phenformin treatment in Figure 2. Interestingly, the results of Annexin V/PI and Brdu staining after phenformin treatment in Figure 5, 7 were different. How to explain these conflicting results?
6. A typo error "exhibite3d" was found at Page 7 line 16.
7. As the authors shown, Lactate/Pyruvate ratio was increased treated by IFN γ (Figure 3A, B), it's confusing that IFN γ did not appreciably alter ECAR (Figure 3C). How to explain this?
8. ROS should be measured after PolyIC treatment in Figure 4.
9. As the authors discussed "NQO1 is recognized as an attractive target in cancer as it is frequently overexpressed in tumors, correlating with late-stage disease and worse survival", it will be much better if the authors can test the cell viability in the NQO1 over-expressed cells in Figure 6 to support their conclusion.
10. To strengthen the conclusion that NQO1 is a bona fide STAT1 target gene that is repressed in response to IFN γ stimulation, a ChIP for STAT1 on the predicted consensus STAT1 binding sites should be performed for +/- IFN γ conditions to confirm the binding of STAT1 on these sites.

February 22, 2021

Dear Reviewers,

We have carefully considered the reviewers' comments pertaining to our manuscript entitled "**STAT1 potentiates oxidative stress revealing a targetable vulnerability that increases phenformin efficacy in breast cancer.**" We thank you for your constructive feedback and feel that their suggestions have considerably strengthened this manuscript. Indeed, this manuscript now contains 9 main figures and 9 supplemental figures. We have included 27 panels of new experimental data to address the reviewers' concerns. Below, we present a detailed rebuttal to each of comment.

REVIEWER #1: We are grateful for this reviewer's positive feedback, who found that, "Overall, the findings reported here have important translational implications, however the biochemical/metabolic mechanism underlying the sensitizing effect of the IFN γ /STAT1/NQO-1 pathway has not been sufficiently worked out." We have performed additional experiments to address the concerns outlined below.

Specific Comment #1. Manipulation of ShcA is predicted to affect several signaling/transcription events beyond STAT1 (including STAT3 inhibition as the authors have previously shown, PMID: 28276425), which can explain changes in the sensitivity of ShcA Y313F cells to phenformin. The authors should provide a stronger rationale for the focus on STAT1 in this system. This is relevant given the importance of STAT3 in metastatic capacity, immune tolerance and rewiring fuel metabolism in breast cancer (PMID: 30100196, 29249690), as well as its capacity to activate antioxidant gene expression (PMID: 21715323). Inhibition of STAT3 will be consistent with many of the cellular phenotype the authors observe.

Response: The reviewer is correct that ShcA regulates both STAT1 and STAT3 signaling in breast cancer cells, as we have previously shown (Ahn et al., Nature Communications, 2017: doi: 10.1038/ncomms14638). In this previous manuscript, we showed that the Y239/240 phosphorylation sites of ShcA increase STAT3 signaling and breast cancer cells expressing ShcA alleles harbouring Y239/240F mutations in both endogenous alleles (Shc2F) show decreased STAT3-Y705 phosphorylation. In contrast, the ShcA Y313 phospho-site

Reviewer Figure #1: Breast cancer cell lines established from MT/ShcA^{+/+} (4788), MT/Shc^{2F/2F} (5372) and MT/Shc^{313F/313F} (6738) transgenic mice were treated with 0.5 mM phenformin or PBS control for 48 hours and viable cells were quantified by trypan blue exclusion. The data is shown as fold change in viability compared to PBS control and is representative of three independent experiments (mean of means \pm SEM). Statistical analysis was

attenuates downstream STAT1 activation such that breast cancer cells expressing two Y313F non-phosphorylatable ShcA alleles show increased total STAT1 and phospho-Y701 STAT1 levels. Our research shows that the ShcA-313 phosphorylation site **does not** decrease STAT3 activation. To address the reviewer's concern, we tested the relative sensitivity of representative breast cancer cell lines expressing two wild-type ShcA alleles as well as those that are homozygous for the Shc2F and Shc313F mutants. We show that both MT/ShcA^{+/+} and MT/Shc^{2F/2F} cells are more resistant to the anti-tumorigenic effects of phenformin relative to MT/Shc^{313F/313F} cells, which are uniquely sensitive to this biguanide (**Reviewer Figure #1**). Thus, at least in MT-transformed breast cancer cells, inhibition of STAT3 signaling is not sufficient to confer phenformin sensitivity. This research does not discount the possibility that STAT3 inhibition could also potentiate the cytotoxic effects of biguanides in other malignances, given the importance of this transcription factor in metabolic reprogramming and immune evasion. Moreover, our data does not formally exclude the possibility that increased phenformin sensitivity in MT/Shc313F cells is also controlled by STAT1-independent mechanisms. However, we respectfully suggest that this is not the goal of the current manuscript and extends beyond the scope of our study. Rather we leveraged our data in MT/Shc313F-expressing cell lines to implicate STAT1 activation in phenformin sensitivity. In support of this data, we show that STAT1 deletion by Crispr/Cas9 genomic editing in MT-transformed cell lines reverses phenformin sensitivity. Moreover, IFN γ stimulation, which activates STAT1 signaling, potentiates the cytotoxic effects of phenformin in 3 independent murine breast cancer cell lines and 8 human breast cancer cell lines.

Specific Comment #2. The spectrum of inflammatory signaling induced by PolyIC is much broader than STAT1/INF γ . It would be important to provide evidence that the observed anti-tumor effect Fig. 2C is specifically mediated by STAT1. A strong direct test for the role of STAT1 in the proposed mechanism of *in vivo* sensitization to phenformin is to test tumor formation in STAT1 KO mice injected as in Fig. 2A or 2C.

Response: The reviewer raised an excellent point. In order to directly test whether STAT1 expression in breast cancer cells is required for the observed synergy between polyIC and phenformin in eliciting a cytotoxic response, we injected MT-transformed breast cancer cells that were either STAT1 wild-type (WT) or STAT1 knock out (KO – by Crispr/Cas9). At 150 mm³, mice were randomized into four groups and treated with polyIC, phenformin, polyIC/phenformin in combination or vehicle control. We show that only STAT1-WT breast cancer cells show increased sensitivity to polyIC/phenformin combination treatment. In contrast, the growth rate of STAT1-deficient tumors, following polyIC or phenformin treatment, alone or in combination, was comparable to that of tumor treated with vehicle control. This data provides direct experimental evidence that breast cancer cells require STAT1 signaling to potentiate the cytotoxic effects of polyIC, together with phenformin. This data is presented below and can also be found in Figure 2d and e. Finally, although we cannot exclude a potential role for STAT1 signaling in cell types within the tumor microenvironment in contributing to the observed synergy with this combination treatment, we focused our attention on intra-tumoral STAT1 function given that we also observed cytotoxic effects with IFN γ /phenformin treatment of breast cancer cell lines *in vitro* and in a STAT1-dependent manner (see Figure 1).

See Figure 2: Panels D and E

Specific Comment #3. It will be important to perform the metabolic characterization in the context of IFN γ and phenformin in addition to IFN γ -only treatment. This is relevant for a deeper understanding of the metabolic understanding of IFN γ 's sensitizing effects.

Response: Based on the fact that phenformin is a potent inhibitor of mitochondrial metabolism, we reasoned that IFN γ was unlikely to further perturb the bioenergetics capacity of breast cancer cells. However, the reviewer is correct that we cannot directly make these conclusions without testing the effects of phenformin, alone or in combination with IFN γ , in altering tumor cell metabolism. We have performed a detailed metabolomics analysis of MT4788 cells treated with IFN γ , phenformin, alone or together, compared to PBS controls. The data is now included in **Figure 3** and is representative of three independent experiments. In addition, in **Supplementary Figure 3**, we show similar metabolomics characterization of independently treated STAT1 wild-type and STAT1 knockout cells to examine whether STAT1 depletion alters the metabolic state of breast cancer cells in response to IFN γ treatment. As expected from the literature, phenformin obliterates the oxygen consumption rate in MT4788 cells (Figure 3a). Whereas IFN γ induces a modest and STAT1-dependent inhibition in the basal and maximal respiration rate of breast cancer cells (see Figure 3B and C along with Figure S3b), IFN γ co-stimulation did not further reduce the exceedingly low OCR in combination with phenformin. Similarly, and as expected, phenformin increased the extracellular acidification rate of breast cancer cells, either alone or in combination with IFN γ , owing to increased glycolytic flux in response to this biguanide (Figure 3i). Again, IFN γ had no impact on phenformin-induced increases in ECAR measurements, irrespective of intra-tumoral STAT1 levels (see Figure 3i and Figure S3d). Moreover, IFN γ lead to a modest (1.5 fold), yet statistically significant decrease in the bioenergetic capacity of breast cancer cells, which was STAT1 dependent. Having said this, phenformin, even as a monotherapy resulted in a 13.4 fold reduction in this parameter, which was not further

decreased following IFN γ co-stimulation (Figure 3H, Figure S3C). Despite the modest effect on IFN γ treatment alone on cellular respiration, it had no impact on the bioenergetic flexibility of breast cancer cells (ie. the amount of ATP produced by oxidative phosphorylation versus glycolysis), either alone or in combination with phenformin. Similarly, STAT1 deletion in breast cancer cells did not alter the bioenergetic flexibility of breast cancer cells (Figure 3k, Figure S3f). We append below data elements that exemplify these data but refer the reviewer to Figures 3 and Supplementary Figure 3 for a complete metabolomic characterization. Taken together, we are confident in our conclusion that phenformin already potently induces energetic stress on breast cancer cells while IFN γ has a significantly more modest effect on cellular bioenergetics. Rather, these data further support our conclusion that the increased cytotoxic effects observed with IFN γ and phenformin on breast tumor growth are likely the result of a combinatorial effect of phenformin-induced energetic stress and IFN γ /STAT1-enhanced oxidative stress (see Figure 4 and Figure 9).

Specific Comment #4. Metabolic profiling of IFN γ -treated cells in figure 3 requires additional depth. The authors conclude IFN γ has minimal effect on bioenergetic parameters, however, changes in fuel metabolism and redox balance can still be relevant. Changes in steady state levels of certain metabolites suggest multiple possibilities some of which are described in the text but these should be directly tested. This is relevant given the extent of published literature on metabolic characteristics of breast cancer and will help advance the field on metabolic mechanisms underlying the sensitizing effect of IFN γ /inflammation on phenformin tumoricidal capacity.

Response: The reviewer is correct that IFN γ /STAT1 signaling may perturb the metabolic state of breast cancer cells, beyond its effects on cellular bioenergetics and may further impact the synthesis of key metabolites, together with phenformin. To test this directly, we also measured steady state levels of key metabolites involved in glycolysis and oxidative phosphorylation, by GC/MS, on breast cancer cells treated with IFN γ and phenformin, alone or in combination, compared to PBS controls.

Again, the data is representative of three independent experiments and can be found in Figure 3 and Supplementary Figure 3 and is appended here also. These studies confirm our original observation that IFN γ treatment alone modestly increased steady state

glycolytic intermediates (pyruvate and lactate) in breast cancer cells and in a STAT1-dependent manner (~1.3 fold) (Figure 3l, S3g). As expected, phenformin alone lead to a profound increase in the levels of these glycolytic intermediates (2.0-2.3 fold), which was modestly further increased in cells co-stimulated with IFN γ and phenformin (2.7-3.0 fold) (Figure 3l). However, only phenformin was able to appreciably increase the lactate/pyruvate ratio in breast cancer cells, which was not further affected by IFN γ co-treatment (Figure 3m). To further assess whether IFN γ signaling altered the viability of breast cancer cells in response to glucose deprivation, we cultured MT4788 cells with either PBS or IFN γ in media containing 25 mM, 5 mM and 0 mM glucose. We show that IFN γ signaling did not induce significant changes in cell viability in response to decreasing glucose levels (Figure S3i). Together, these data demonstrate that IFN γ minimally impacts glucose metabolism, either alone or in combination with phenformin.

We performed similar studies examining whether IFN γ altered mitochondrial metabolism in breast cancer cells. Indeed, IFN γ treatment modestly increased α -ketoglutarate levels (1.5 fold) and in a STAT1-dependent manner (Figure 3n, S3j). In contrast, phenformin lead to a more robust (4.7 fold) increase in α -ketoglutarate levels and combined IFN γ /phenformin treatment led to a further increase (5.8 fold) in α -ketoglutarate levels, which was statistically significant (Figure 3n). The observation that IFN γ may induce reductive carboxylation of glutamine metabolism is supported by the fact that IFN γ treatment alone lead to a 1.4-1.6 fold increase in the α -ketoglutarate/citrate ratio, and in a STAT1-dependent manner (Figure 3k). Moreover IFN γ further increased the α -ketoglutarate/citrate ratio in response to phenformin treatment (24 fold with phenformin alone compared to 31 fold with IFN γ /phenformin treatment (Figure 3O). We also measured the ability of IFN γ to modulate cell viability in response to glutamine withdrawal (4 mM, 0.4 mM, 0 mM) and show that IFN γ treatment actually increases breast cancer cell viability in response to a 10 fold reduction in glutamine levels in the media. This protective effect was lost in the absence of glutamine (Figure S3l). These observations suggest that IFN γ may actually increase glutamine metabolism, to favor reductive carboxylation. While interesting, this new knowledge is beyond the scope of this study. Of relevance to this manuscript, IFN γ stimulation actually **protected** cells from glutamine withdrawal, suggesting that the ability of this cytokine to alter glutamine metabolism cannot explain the increased cytotoxicity of combined IFN γ /phenformin treatment. Although glutamine is essential for redox balance by contributing to glutathione production, we show that IFN γ signaling does not appreciably alter glutathione levels in MT4788 breast cancer cells (Supplementary Figure 4). Taken together, our data does not support the hypothesis that metabolic rewiring by IFN γ contributes to perturbation of redox balance.

Specific Comment #5. The insight gained from ECAR data in Fig. 3C is unclear. How do the authors interpret the ECAR data vis-à-vis changes in lactate and pyruvate.

Response: As outlined above, the effects of IFN γ signaling on glucose metabolism are quite modest compared to what is observed with phenformin (as measured by pyruvate and lactate levels as well as the lactate/pyruvate ratio). Indeed, the reviewer is correct

that we do not observe a similar increase in ECAR levels with IFN γ treatment alone. However, we do show that phenformin leads to large increases in ECAR levels, as expected from the literature (Figure 3i). This data likely suggests that the relatively modest increases in intracellular lactate levels produced in response to IFN γ are not being exported from the cell. This is consistent with the fact that IFN γ treatment does not actually increase the lactate/pyruvate ratio in MT4788, unlike phenformin which does increase this ratio (Figure 3m).

Specific Comment #6. Overall, the main figures will benefit from some reorganization to more effectively package key data and eliminate figure panels that are not absolutely relevant to the main discovery of the paper. Examples include (but are not limited to) Fig. 1G and H, which can be moved to supplements.

Response: We have significantly re-organized the figures and moved several panels into the supplementary figures, highlighting only those data elements that are essential to support the conclusion of this study in the main figures. We hope the reviewer feels that the re-structuring of the figures is improved in the revised manuscript.

REVIEWER #2: We are grateful for this reviewer's positive feedback, who stated that, "This paper is characterized by an abundance of experimental data in a wide variety of systems." However, several points were raised, which we have addressed:

Specific Comment #1. There are, in fact, so many different systems used with unclear rationales, that it is difficult to crystallize specific high-impact findings. For example, data are presented on combinations of phenformin with an oncolytic virus or with an anti-PD-1 antibody, which have almost nothing to do with the rest of the data, and the material on b-lapachone and IACS-010759 is similarly very tangential. A tighter focus on the effects of STAT1 on NQO1, and how this synergizes with phenformin (or other related compounds), as indicated by the title of the manuscript, would likely be much more effective.

Response: We thank the reviewer for this feedback and we have significantly re-structured the manuscript. Please see our response to comment #6 from reviewer 1. Regarding the specific concerns raised here, we respectfully disagree that the data with the oncolytic virus or anti-PD1 blockade is tangential. Indeed, many studies focus on the ability of STAT1 to induce anti-tumor immune responses, given the well-documented role of this transcription factor in promoting immune surveillance. Given this fact, and coupled with studies suggesting that biguanides may also affect the function of immune cells (see last paragraph of the introduction), we felt it important to experimentally address whether the ability of IFN γ /STAT1 signaling to collaborate with phenformin in invoking tumoricidal responses was, in part, immune-mediated. Using genetic approaches (CD8 $^{-/-}$ mice) as well as therapies that potentiate anti-tumor immunity as part of their mechanism of action (PD1 checkpoint blockade; oncolytic virus), we can feel confident in our conclusion that polyIC/phenformin combination therapy does not require

CTL-driven immune responses to potentiate its cytotoxic effects against breast cancer cells. We agree that this observation is not the main conclusion of our studies and is therefore found in the supplemental figures.

By the same token, we retain the data describing the effects of IACS-010759 on breast cancer cell viability in combination with β -lapachone. We agree with the reviewer that a major finding of our study is the ability of IFN γ /STAT1 signaling to decrease the expression of NQO1, reducing levels of an important ROS scavenger in breast cancer cells and thereby facilitating phenformin-induced oxidative stress. The combination therapies involving β -lapachone and phenformin or IACS-010759 further re-inforce these findings and provide pre-clinical evidence that this approach may deserve clinical translation. Biguanides and IACS-010659 inhibit mitochondrial complex I by distinct mechanisms (PMID: 25017630, PMID: 29892070). Finally, a recent report indicated that phenformin may also inhibit mitochondrial metabolism, independently of complex I, by inhibiting glycerophosphate dehydrogenase activity, leading to elevated NADH levels (PMID: 32049007). By showing that both phenformin and IACS-010759 can collaborate with β -lapachone to decrease breast cancer cell viability, these data further support our conclusion that complex I inhibition underlies the increased cytotoxicity associated with this NQO1 bioactivatable drug.

Specific Comment #2. In the sections on the role of interferon-g and STAT1 in mediating effects with phenformin (and NQO1), the concepts of increased STAT1 expression, STAT1 phosphorylation, and STAT1 transcriptional activity are used intermittently, without critically assessing their relationships to each other. Since transcriptionally active STAT1 can increase STAT1 transcription, increased STAT1 levels can reflect increased transcriptional activity. However, it is not clear if the authors are trying to make this point. Similarly, STAT1 is thought to be transcriptionally active only when phosphorylated on tyrosine-701, though counter-examples have been proposed. Since these are critical points to understand the connection with phenformin sensitivity and NQO1 repression, they should be addressed directly

Response: The reviewer raises an excellent point. First, using STAT1 knock out breast cancer cells (engineered by Crispr/Cas9), we provide function evidence that STAT1 expression in breast cancer cells is required to promote IFN γ /phenformin sensitivity *in vitro* (Figure 1e) as well as the anti-tumorigenic effects of polyIC/phenformin treatment *in vivo* (Figure 2d and e; see response to reviewer #1; specific comment #2). In addition, we further show that the ability of IFN γ to reduce Nqo1 mRNA expression levels requires STAT1 in murine MT4788 breast cancer cells (Figure S6; Figure S7a; see figure below). In order to better understand how STAT1 controls NQO1 expression, we extended our characterization to multiple breast cancer cell lines across distinct subtypes: ER+/HER2-

**Steady State Levels
(Figure 6B)**

**IFN γ stimulation activates STAT1 but has no effect on NQO1 mRNA levels
in human breast cancer cells (Figure S7C)**

**IFN γ stimulation activates STAT1 and reduces NQO1 protein levels
in human breast cancer cells (Figure 6C)**

**Mouse breast cancer cells
(Figure S7A)**

(MCF7), ER+/HER2+ (BT474); ER-/HER2+ (HCC1954) and ER-/HER2- (MDA-MB-231, BT20, MDA-MB-436, Hs578T and BT549). Importantly, these cells differ in their STAT1 and pY701-STAT1 levels under basal growth conditions. NQO1 immunoblot analysis showed that at steady state, neither STAT1 nor pY701-STAT1 levels correlate with relative NQO1 expression levels (see Figure 6b; quantification of the immunoblots is shown in Figure 6). We next tested whether IFN γ stimulation impacted NQO1 levels across these cell lines. As expected IFN γ treatment increased Y701-STAT1 phosphorylation across all 8 cell lines (see above and Figure 6b). We also observed significantly reduced NQO1 protein levels following IFN γ stimulation in three TNBC cell lines (MDA-MB-231, Hs578T, MDA-MB-436 – see above and Figure 6c). This data is representative of three independent experiments. Therefore, IFN γ stimulation is able to reduce NQO1 expression in some but not all breast cancer cells, indicating multiple levels of gene expression. This is also consistent with our data showing a trend between higher STAT1 levels and lower NQO1 protein levels in patient derived xenografts from TNBC and lung cancer patients (Figure S7d and Figure S9). However, we were surprised to find that *NQO1* mRNA levels were not reduced in any of the cell lines tested in response to IFN γ treatment. This is in comparison to a bona fide CXCL9, which is a bona fide transcriptional target for STAT1 (see above and Figure S7c). Therefore, although we identified NQO1 as a putative STAT1 target gene by RNAseq (comparing IFN γ -stimulated STAT1-WT and STAT1-KO MT4788 cells – Figure S6), which we validated by RT-qPCR analysis (see above and Figure S7a), these data suggest that IFN γ inhibition of NQO1 expression in a panel of human breast cancer cells is likely to be regulated post-transcriptionally.

In light of these observations, we removed the data from the earlier version of the manuscript, suggesting that the human and mouse promoters of NQO1 contain consensus STAT1 binding sites and that both human and mouse NQO1 mRNA levels

were found to be repressed following IFN γ treatment in independent cells (see below: original data can be found at www.interferome.org). We also removed the data in the original manuscript looking at relative *Nqo1* mRNA expression levels in both primary breast and lung cancers (from the TCGA). Recall, that in this analysis, we employed a transcriptional signature of known STAT1 target genes (STAT1 ssGSEA) to rank order tumors (the higher the ssGSEA score denotes increased expression of STAT1-regulated target genes). In our original submission, we showed no correlation between relative *Nqo1* levels and the STAT1 ssGSEA score in primary breast cancers. However, we did observe a statistically significant inverse correlation between *Nqo1* levels and the STAT1 ssGSEA score in primary lung cancers (see below and Figure S5 of our original submission). Therefore, we cannot exclude the possibility that IFN γ -stimulated STAT1 activation does lead to transcriptional repression of *NQO1* in a subset of cancers, although our data does not support that this is a major mechanism in human breast cancers. Rather, our new data suggests that the mechanism of action is likely to be post-transcriptional. While interesting, elucidating the mechanism(s) by which STAT1 controls *NQO1* gene expression will require future research but is beyond the scope of this study. Rather, by showing that STAT1 can repress *NQO1* gene expression, this research provides the first evidence that STAT1 may potentiate oxidative stress by decreasing the ROS scavenging potential in cancer cells, exposing a novel therapeutic vulnerability in combination with complex I inhibitors, which increase ROS levels. Finally, our research also suggests that IFN γ -induced oxidative stress relies on additional mechanisms that include but extend beyond regulation of *NQO1* expression. Indeed, five of the human breast cancer cell lines are sensitive to combined IFN γ /phenformin treatment (MCF7, HCC1954, BT474, BT20, BT549) yet do not show evidence for reduced *NQO1* protein levels following IFN γ treatment (compare Figure 1c and Figure 6c). Of these BT474 cells are exquisitely sensitive to IFN γ /phenformin treatment and these cytotoxic effects can be completely reversed with a mitochondrial ROS scavenger (MitoTEMPO, Figure 4e). While interesting, identification of the additional mechanisms by which IFN γ potentiates oxidative damage is beyond the scope of this study.

INTERFEROME (NQO1 PROMOTER)

TCGA ANALYSIS

Specific Comment #3. The second section of the results concerns the effect of “inflammation” on sensitizing breast cancer models to the tumoricidal effects of phenformin. While this seems to relate to findings with Poly IC, it is not clear how the authors are defining inflammation, as there are many contradictory findings here. In fact, the authors state that the “observed cooperativity is likely distinct from metabolic stress or inflammatory processes.” A subsequent section is entitled, “Inflammation-induced phenformin sensitivity requires mitochondrial ROS.” Here, inflammation seems to be referring to IFN-g treatment (which is likely not “inflammatory” in these systems in any conventional sense). Even the term “STAT1-induced inflammation” in the title seems imprecise given the data in the paper (and it is not clear that STAT1 can “induce” inflammation). This section in particular, and the manuscript in general, would benefit from more precision in the use of scientific terms.

Response: We agree entirely with the reviewer. We instead refer to IFN γ /STAT1 activation in the title as opposed to any inflammatory response that this signaling pathway may elicit. We have also modified the title of the manuscript accordingly to reflect this change. The new title is now “*STAT1 potentiates oxidative stress revealing a targetable vulnerability that increases phenformin efficacy in breast cancer*”

Specific Comment #4. The issue of the known human pharmacokinetics of these drugs should be addressed. One of the reasons that metformin (and potentially phenformin) have not performed as well in human studies as in cell culture and murine studies is that the therapeutic levels achieved in people are much lower. This issue should at least be noted in this manuscript.

Response: We agree entirely with the reviewer and this point was addressed in the original manuscript in the introduction. The text can be found in the last paragraph on page 3 of the current manuscript and reads as follows:

“The lack of durable responses with biguanides in clinical trials can be explained, in part, by the fact that pre-clinical studies use significantly higher drug concentrations than can be achieved clinically^{14, 15}. Moreover, metabolic flexibility and nutrient availability in the tumor microenvironment may contribute to the poor efficacy of biguanides as single anti-cancer agents^{16, 17, 18}. Rational combination strategies that lower the concentrations of biguanides needed for anti-neoplastic activity may revitalize the therapeutic potential of this drug class in oncology¹⁹.”

Taking this point into consideration, recall our data showing that polyIC treatment confers comparable tumoricidal effects with a 5 fold lower dose of phenformin (10 mg/kg vs 50 mg/kg phenformin – see Figure 2j).

REVIEWER #3: We are grateful for this reviewer’s positive feedback, who stated that, “The findings are very interesting and will provide the important knowledge to design new drug strategy in oncology. Although the authors have presented the extensive data,

a number of concerns given below still need to be addressed to improve the whole manuscript.” We have addressed each of these concerns as follows:

Specific Comment #1. In the Western blot data for some cell lines, it looks like IFN γ -STAT1 has regulation effects on Tubulin expression. The authors may need to be careful about this internal control pick. For example, the expression of Tubulin reduced in 6738 cells that were stimulated by IFN γ (see Figure 1B) and in MT4788 cells with STAT1 KO (see Figure 1G).

Response: We performed new immunoblots and show comparable Tubulin expression levels among the various conditions (now Figure 1d, Figure 6c).

Specific Comment #2. In Figure 1, when the authors tested the cooperative effects of IFN γ and phenformin in different breast cancer cell lines, they did not test any ER+ cell lines such as MCF7. Can the authors give me an explanation for not doing tests in ER+ breast cancer lines?

Response: In the original submission, we included BT474 cells, which are an ER+/HER2+ breast cancer cell line. However, the author is correct that we did not test ER+/HER2- cells. We have therefore expanded our analyses to 8 human breast cancer

IFN γ /phenformin combination treatment decreases the viability of multiple breast cancer cell lines, irrespective of subtype (Figure 1c).

cell lines across all subtypes, including (1) ER+/HER2- (MCF7), (2) ER+/HER2+ (BT474), (3) ER-/HER2+ (HCC1954) and (4) TNBC (MDA-MB-231, BT20, MDA-MB-436, Hs578T and BT549). We show that IFN γ stimulation potentiates phenformin sensitivity across all cell lines tested (see above and Figure 1c).

Specific Comment #3. The data in Figure 1I panel are showing some conflict: IFN- γ treatment alone had no effect on cell viability of MT4788 (STAT1-WT). But IFN- γ treatment alone increases cell viability of MT4788 (STAT1-KO)? An explanation is needed for this phenomenon.

Response: Figure 1i is now figure 1e of the current manuscript. In both MT864 and MT4788 cells, IFN γ stimulation alone had no impact on cell viability (compared to PBS controls), both in STAT1-wild-type and STAT1-knockout cells. The issue with the previous figure is that we originally normalized the number of cells to the PBS-treated, STAT1 wild-type controls. Whereas STAT1 deletion had no impact on the growth of MT864 cells, we did observe a 1.3 fold increase in the growth of MT4788 cells compared to their parental controls. This likely reflects differences in clonal variability when we pooled the STAT1-KO cells. Thus, MT4788-KO cells (PBS and IFN γ -stimulated) showed an increase cell number compared to MT4788-WT cells in the original figure; however, there were no differences in cell number *between* PBS and IFN γ -stimulated MT4788-KO cells. We thank the reviewer for point out this discrepancy. We have now normalized the drug-treated cells (STAT1-WT and STAT1-KO) relative to each of their own respective PBS controls, to more properly interpret the effects of phenformin and IFN γ on cell viability *within* individual cell lines (as we did for all other *in vitro* growth curves). This re-plotted data can be found below and is figure 1e of the current manuscript.

Specific Comment #4. For some reason, when the authors collected the tumor volume data under different treatments for the different panels in Figure S1 and Figure 2, they used different termination days (8, 10 and 13) for different panel. I feel the results were completely influenced by the termination time pick. For example, why the authors chose DAY10 not DAY8 in Figure S1A but DAY8 in Figure 2A-C? I believe the different timing pick will cause bias to make the right conclusion. For example, I can conclude that there is no obviously additive effect on DAY8 in Figure S1A. Moreover, Figure S1B missed the data on DAY8 for some reason (it could be explained as the PFU of oncolytic virus was too high so that the tumor progression was almost completely suppressed at the first 6 days of treatment and no further effect in the combination group at last). There should be an additive effect in the combination group because the mechanisms of oncolytic virus and Phenformin must be different.

Response: We appreciate the reviewer's concern but would like to point out that the experimental endpoint is based on the time when tumors in the control cohort reached a similar tumor volume (~500 mm³). Given the variability that is inherent to *in vivo* studies, this can vary somewhat between individual experiments but we do not feel that a few days difference in the experimental endpoint will profoundly change our interpretation. Indeed, for the MFP data in Figure S1 (now Figure S2), we conclude that **neither** anti-PD1 antibodies **nor** oncolytic virus synergized with phenformin in reducing the tumor growth kinetics and this conclusion would not be different by terminating the experiment a few days sooner.

Specific Comment #5. The connection of the anti-tumor mechanism of phenformin to apoptosis needs to be more careful for some data. There is no difference of Ki67 expression or Casp3 cleavage after phenformin treatment in Figure 2. Interestingly, the results of Annexin V/PI and BrdU staining after phenformin treatment in Figure 5, 7 were different. How to explain these conflicting results?

Response: We ask the reviewer to keep in mind that the immunohistochemical studies on FFPE tumor tissue (Figure 2) compares steady state changes in proliferation (Ki67) or apoptosis (cleaved caspase 3) in a heterogeneous population of cells and at the experimental endpoint. In contrast, the BrdU and Annexin V studies are done on a purified cell population within a 48 hour time point post drug treatment. Despite these more controlled conditions, phenformin treatment alone had relatively modest effects on cell viability *in vitro* (Annexin V staining, Figure 5d and 7b), neither of which reached statistical significance. Moreover, the effects of phenformin alone on cell proliferation *in vitro* (BrdU staining, Figure 5e and 7c) were relatively modest (~15% reduction). Therefore, it is not surprising that these subtle differences are not reflected in the tumors (based on our IHC studies). However, we do observe robust increases in apoptosis with the polyIC/phenformin combination treatment in tumors, which is consistent with a much

larger and appreciable increase apoptosis of cells treated with BSO in combination with phenformin *in vitro*.

Specific Comment #6. A typo error “exhibite3d” was found at Page 7 line 16.

Response: This has been corrected.

Specific Comment #7. As the authors shown, Lactate/Pyruvate ratio was increased treated by IFN γ (Figure 3A, B), it’s confusing that IFN γ did not appreciably alter ECAR (Figure 3C). How to explain this?

Response: We direct the reviewer to our response to Reviewer #1, specific comment #5 for how we addressed this issue.

Specific Comment #8. ROS should be measured after PolyIC treatment in Figure 4.

Response: Reactive oxygen species have an exceedingly short half-life (for example the half-life of superoxide anion is 10^{-5} seconds). The polyIC treatment studies are performed in tumor bearing mice *in vivo*. Given this fact, it would be impossible for us to dissociate tumor cells from drug-treated mice and accurately measure ROS levels *ex vivo* using fluorescent probes. To overcome this limitation, we provided direct evidence that co-administration of a mitochondrial ROS scavenger (MitoTEMPO) *in vivo* reverses the tumoricidal effects of polyIC/phenformin treatment (Figure 4f). Combined with the fact that we also observe increased oxidative DNA damage (8-oxodG) in tumors treated with this drug combination (Figure 4g), these data provide compelling evidence that polyIC collaborates with phenformin to induce oxidative stress.

Specific Comment #9. As the authors discussed “NQO1 is recognized as an attractive target in cancer as it is frequently overexpressed in tumors, correlating with late-stage disease and worse survival”, it will be much better if the authors can test the cell viability in the NQO1 over-expressed cells in Figure 6 to support their conclusion.

Response: Several studies have shown an association between elevated NQO1 expression levels and poor outcome, which are highlighted in our manuscript. Addressing whether or not NQO1 overexpression on its own impacts the growth potential breast cancer cells is not the aim of this study. In fact, under non-stressed conditions, it is possible that NQO1 overexpression will not appreciably alter cell growth. Rather, as NQO1 is a ROS scavenger, we propose that NQO1 overexpression will increase cell viability in response to stimuli that induce oxidative stress. These are multi-factorial and can include biological states such as hypoxia or therapy-induced stressors, including chemotherapy or in this case, biguanides. The ability of elevated NQO1 levels to promote tumor survival, leading to more aggressive disease, is therefore likely multi-

factorial, but is not the main point of the paper. Rather, we show that inhibition of NQO1 via genetic or pharmacological approaches (β -lapachone) may represent a viable treatment strategy for women with NQO1-positive breast tumors.

Specific Comment #10. To strengthen the conclusion that NQO1 is a bona fide STAT1 target gene that is repressed in response to IFN γ stimulation, a ChIP for STAT1 on the predicted consensus STAT1 binding sites should be performed for +/- IFN γ conditions to confirm the binding of STAT1 on these sites.

Response: We performed a significant number of studies to characterize how IFN γ controls NQO1 expression in a large panel of breast cancer cell lines and have identified both transcriptional and post-transcriptional mechanisms of action (see response to Reviewer #2; Specific Comment #2). To summarize briefly, we show that while IFN γ stimulation decreases NQO1 mRNA levels in MT4788 breast cancer cells (and in a STAT1-dependent manner), IFN γ -induced repression of NQO1 expression in several human breast cancer cell lines is regulated at the post-transcriptional level (including MDA-MB-231 cells). Moreover, we performed STAT1-CHIP experiments in both MT4788 and MDA-MB-231 cells looking at two time points following IFN γ treatment (1h and 24h). We measured relative STAT1 binding to the promoter or transcriptional start site (TSS) of NQO1 (along with intron 1 as a negative control). As a positive control, we include the promoter and TSS of IRF1, which is a known STAT1 target gene. While IRF1 showed STAT1 binding, we did not observe STAT1 binding to the promoter or TSS of either breast cancer cell line. Thus, we cannot conclude that NQO1 is a direct STAT1 target gene and we removed the statements alluding to this from the manuscript. Rather, IFN γ /STAT1 signaling indirectly controls NQO1 gene expression through complex mechanisms. While interesting to decipher, this would involve a new line of investigation and is not the point of this study.

Again, we thank each of the reviewers for their valuable input. We feel that we have adequately addressed their critiques and that the revised manuscript has significantly improved. We feel that this work provides the novelty and biological relevance required for publication in *Nature Communications*. We look forward to hearing from you soon.

REVIEWER COMMENTS

Reviewer #1 (Remarks to the Author):

The authors have adequately addressed my previous comments.

Reviewer #2 (Remarks to the Author):

The manuscript has been marginally improved. It still contains a great deal of data that are not completely coherent with regard to the underlying theme, and the concentrations of the biguanides seem to significantly exceed therapeutic levels in patients. However, overall it does contain interesting information on the intersection between interferon effects and biguanides.

The writing could still be more rigorous. For example, line 433: "Resistant cancer cells must therefore arm themselves by increasing their anti-oxidant defenses." Cells do not arm themselves, although cancer cells with enhanced ability to detoxify ROS may have a survival advantage under the selection pressure of therapy-induced ROS production.

Reviewer #3 (Remarks to the Author):

Overall, I think the authors did a good job addressing all the concerns that I have raised during my first review and this manuscript has been greatly approved.

We have carefully considered the remaining comments from reviewer #2 pertaining to our manuscript entitled “**STAT1 potentiates oxidative stress revealing a targetable vulnerability that increases phenformin efficacy in breast cancer.**” They have been addressed in this revised manuscript as follows:

Reviewer Comment: The manuscript has been marginally improved. It still contains a great deal of data that are not completely coherent with regard to the underlying theme, and the concentrations of the biguanides seem to significantly exceed therapeutic levels in patients. However, overall it does contain interesting information on the intersection between interferon effects and biguanides.

Response: We have addressed the issue regarding concentrations of biguanides employed in this study, by incorporating the following text into the discussion:

The higher drug concentrations of biguanides required to elicit anti-neoplastic effects in pre-clinical studies has impeded our ability to translate findings in the treatment of cancer patients^{14, 15}. Indeed, pharmacokinetic studies demonstrated that diabetic patients treated with metformin achieve maximal plasma concentrations that are at least 6-10 fold less than what is observed in pre-clinical cancer models treated with metformin^{14, 15}. Notwithstanding the fact that similar pharmacokinetic studies have yet to be performed with phenformin or other complex I inhibitors, we showed that polyIC co-treatment elicited comparable anti-neoplastic effects with 5-10 fold lower doses of phenformin (10 mg/kg) than what is typically used in pre-clinical studies in oncology (50-100 mg/kg). Clinical trials are required to confirm these studies in cancer patients.

Reviewer Comment: The writing could still be more rigorous. For example, line 433: “Resistant cancer cells must therefore arm themselves by increasing their anti-oxidant defenses.” Cells do not arm themselves, although cancer cells with enhanced ability to detoxify ROS may have a survival advantage under the selection pressure of therapy-induced ROS production.

Response: We have carefully reviewed our text for clarity and made the necessary modifications. All changes are highlighted in the marked up version of your manuscript.

We thank you for your feedback and efforts on our behalf. We feel that the revised manuscript is suitable for publication in *Nature Communications*.